# GWAS on short tandem repeats identifies genetic mechanisms in Alzheimer's disease

David Gmelin[1], Olena Ohlei [1,2], M. Muaaz Aslam[1], Marit P. Junge [1],
Laura Parkkinen [3], Kristina Mullin[4], Dmitry Prokopenko [4,5],
Christina M. Lill [2,6], Rudolph E. Tanzi [4,5], Valerija Dobricic[1,7] &
Lars Bertram [1,7] ✉

GWAS typically focus on SNPs, often excluding complex genetic variants, such as short tandem repeats. Here, we report the results of GWAS analyses systematically assessing the role of short tandem repeats, both imputed and directly genotyped by whole genome sequencing, on risk for Alzheimer's disease in a large collection of ~330,000 individuals (3287 cases; 47,048 Alzheimer's disease-by-proxy cases, 283,111 controls) from the UK biobank. Using short tandem repeat genotype data, we identify 15 independent loci showing evidence for genome-wide significant association with Alzheimer's disease risk. While most identified loci had already been highlighted by SNP-based GWAS, we detect short tandem repeat-based signals near the genes *SNX32* (chr. 11q13) and *WSB1* (chr. 17q11). In addition, we delineate several other loci where short tandem repeats (and not SNPs) either represent the lead signal (*ABCA7*) or make substantial contributions to the SNP-driven associations (*HLA-DRB1, MINDY/ADAM10*, and *APOE*). Heritability analyses estimate that short tandem repeats account for at least 3% of the total phenotypic variance of Alzheimer's disease in this dataset. Aligning our top short tandem repeats with DNA methylation and transcriptome profiles from human brain samples suggests that several short tandem repeats may unfold their effects by impacting gene expression.

Alzheimer's disease (AD) is a genetically complex neurodegenerative disorder. For early-onset familial AD (EOFAD), rare, disease-causing mutations have been identified in three genes (*APP, PSEN1, PSEN2*), typically leading to an onset before 65 years[1]. However, the vast majority of AD cases have a later disease onset (commonly referred to as late-onset AD [LOAD]) genetically governed by a polygenic risk architecture[2]. For this latter form, genome-wide association studies (GWAS) have uncovered common variants (typically single-nucleotide polymorphisms [SNPs]) at more than 70 loci significantly modifying

disease risk[3]. While GWAS have substantially enhanced our understanding of AD genetics, a large portion of the phenotypic variance remains unexplained by the hitherto known loci[4], a situation often referred to as "missing heritability"[5].

A sizeable portion of this missing heritability may be elicited by the effects of other types of genetic variants, such as tandem repeat (TR) polymorphisms. TRs represent repetitive elements of the genome consisting of units of specific nucleotide sequences ("repeats") occurring adjacent to each other in a head-to-tail fashion ("tandem").

[1]Lübeck Interdisciplinary Platform for Genome Analytics (LIGA), University of Lübeck, Lübeck, Germany. [2]Institute of Epidemiology and Social Medicine, University of Münster, Münster, Germany. [3]Nuffield Department of Clinical Neurosciences, University of Oxford, Oxford, UK. [4]Genetics and Aging Research Unit and McCance Center for Brain Health, Department of Neurology, Massachusetts General Hospital, Boston, MA, USA. [5]Harvard Medical School, Boston, MA, USA. [6]Ageing Epidemiology Research Unit (AGE), School of Public Health, Imperial College London, London, UK. [7]These authors contributed equally: Valerija Dobricic, Lars Bertram. ✉e-mail: lars.bertram@uni-luebeck.de

Short tandem repeats (STRs) are a subcategory of TRs where the repeat unit size is between 1 and 6 base pairs (bp)[6]. It is well established that STRs may represent a direct cause of disease through expansions of the TR sequence (e.g., STR expansions causing Huntington's disease or fragile X syndrome[7]). Typically, these TR expansions are very rare and function like monogenic disease-causing mutations. Notwithstanding, the length of most STRs is highly variable, and these length polymorphisms have been associated with gene expression[8], mRNA splicing[9], DNA methylation[10], as well as disease risk[11,12], emphasizing their role in shaping diverse human traits.

Despite their importance, a systematic analysis of STR effects on a genome-wide scale was hindered by technological limitations until recently. This has now changed with the advent of cost-efficient whole genome sequencing (WGS) approaches and the development of TR-aware computational tools[13,14]. Together, these developments have led to the creation of WGS-derived STR reference panels that allow for imputing STR polymorphisms from SNP genotyping data[15,16]. Despite these developments, there are still comparatively few studies that have interrogated the role of STRs in complex traits at a genome-wide scale. These have either used imputed STRs[11,12,17,18] or STRs called directly from WGS data[19,20]. Collectively, the (sparse) literature available on the topic to date suggests that 5–10% of complex trait GWAS signals may be elicited by STRs rather than SNPs[11,18]. For AD, we are only aware of one study on the topic[19]. Perhaps owing to its comparatively small sample size (n = ~3000), the only locus identified to show significant STR-related effects in that study was *APOE*[19]. In summary, STR-based GWAS are still rare for complex traits, including AD. The few studies that have been published to date concluded that STRs likely make an important and hitherto underappreciated contribution to the genetic architecture of complex traits.

Here, we set out to thoroughly and systematically assess the potential role of STR polymorphisms in contributing to AD risk. Overall, we identify 254 imputed STRs across 14 independent loci and one WGS-derived STR to show genome-wide significant association with AD risk in this dataset.

## Results

We performed STR-based GWAS analyses using both imputed and WGS-derived STR data in 333,446 (3287 AD cases; 47,048 AD-by-proxy cases) and 107,289 (1026 AD cases, 15,089 AD-by-proxy cases) samples of the UK Biobank (UKB), respectively. For the main analyses, STRs were analyzed in bi-allelic tests, followed by multi-allelic analyses for variants attaining genome-wide significance in the bi-allelic tests. Next, genome-wide significant loci were assessed in validation analyses using independent subsets within the UKB cohort. Lastly, associated STR loci were subjected to in silico fine-mapping to disentangle the respective roles of SNPs vs. STRs, and to a first-pass functional characterization using brain-based meQTL and eQTL data.

### Genome-wide association analyses on imputed STRs

After quality control (QC), we were able to include a total of 333,446 genetically unrelated UKB participants (of different ancestries; see the "Methods" section), comprising 3287 clinically diagnosed AD cases, 47,048 AD-by-proxy cases, and 283,111 controls (Supplementary Fig. 1 and Supplementary Table 1). For discovery, we performed GWAS analyses on 295,551 individuals self-reported as "White-British" (discovery cohort: 2947 AD cases, 42,923 AD-by-proxy, and 249,681 controls; Supplementary Table 1) and 3,026,404 imputed STR variants that passed QC (see the "Methods" section). These analyses resulted in 254 variants at 14 independent loci that passed our genome-wide significance threshold after multiple testing correction (P < 1.49E−07; see the "Methods" section). In addition, there were 228 variants across 21 independent loci showing at least genome-wide suggestive (alpha = 1.0E−05) evidence of association (Fig. 1A, Supplementary Data 1). Comparison of observed vs. expected genome-wide test statistics

revealed a λ of 1.056, suggesting that our STR-based GWAS results are not affected by undue inflation (Supplementary Fig. 2A).

Most of the detected genome-wide significant STR associations (n = 153 unique STRs) were within the *APOE* region on chromosome 19q13.32, a well-established and likely SNP-based risk factor for AD[21,22]. The remaining 76 STRs mapped to 13 independent loci outside the *APOE* region, most often to the vicinity (i.e. 1.3–138.5 kb distance) of known AD GWAS signals (based on ref. 3). Only one STR signal (i.e. on chr11q13.1) mapped into an intergenic region outside known AD GWAS signals (nearest genes *OVOL1* and *SNX32*; Fig. 1A, Supplementary Fig. 3). In eight of the 13 non-*APOE* loci, STRs were located within the open-reading frame (ORF) of protein coding genes (all intronic), while five were between −5 and 27 kb up-/down-stream from the nearest protein coding gene (Table 1, Supplementary Data 2).

With respect to the 228 suggestive significant STRs, the majority (n = 178, across 21 independent loci) were located outside the *APOE* region (Supplementary Data 3). Thirteen of these peaks are located more than 500 kb from any known AD locus and, hence, may at least partially represent novel AD risk signals. Among these 13, eight are within the ORF (all intronic) while the remaining five STRs are mapped between −2 and 310 kb up-/down-stream from the closest protein-coding gene.

To search for sex-specific differences in the primary GWAS, we performed two types of analyses: First, we ran sex-stratified GWAS analyses, but these did not reveal any novel signals absent from the primary GWAS (Supplementary Fig. 26, Supplementary Data 18). Second, we searched for loci showing substantial association differences in males vs. females using genome-wide interaction analyses, including a 'genotype × sex' interaction term. Again, no variants showed evidence for genome-wide significant interaction (Supplementary Fig. 27). Overall, neither analysis provided strong evidence for sex-specific differences in the primary GWAS signals.

Sensitivity analyses, including age as a covariate, show a strong agreement with the results from the non-age-adjusted analyses. Specifically, Pearson correlation coefficients comparing beta values with and without adjustment are r = 0.9984 for all variants and r = 1 when limiting the comparison to at least suggestively significant variants (both; P < 2.2E−308; Supplementary Fig. 25, Supplementary Data 17). Quantification of the potential "gain in precision" resulting from including age as a covariate (see the "Methods" section) revealed almost no reduction in the SE ratio and, hence, gain in precision (median reduction = 0.16–0.17%). The overall negligible effect of including age in the model is not unexpected since this variable cannot be considered a confounder in this context.

Results above are based on a bi-allelic coding of STRs, which we used owing to a number of limitations with current STR imputation catalogs requiring extensive manual curation, which was beyond the scope of this study (see the "Methods" section). However, to exclude the possibility that some of our bi-allelic signals represent false-positive associations, we performed targeted revisions of the imputation catalog to allow multi-allelic analyses for all STRs ±250 kb around our suggestive and genome-wide significant hits (see the "Methods" section; Supplementary Data 4, Supplementary Fig. 4). These analyses revealed that 12 out of 14 top-hits remained genome-wide significant with only negligible changes of P values, but two signals (both in close physical proximity on chr. 15, i.e. 15:58846596 and 15:63139195) no longer showed genome-wide significant associations after applying the multi-allelic approach (Supplementary Data 4). However, we did identify other nearby STRs within these chr. 15 loci both eliciting genome-wide significant signals, i.e. 15:58915192, P = 2.69E−08 and 15:63233055, P = 1.30E−07, providing support for the robustness of these signals. Interestingly, we also found three STRs within the ±250 kb window that show more significant associations than the initial bi-allelic top hits (i.e., 6:32472900 vs. 6:32611487, 11:60251788 vs. 11:60177578, and 17:63469860 vs. 17:63485772;

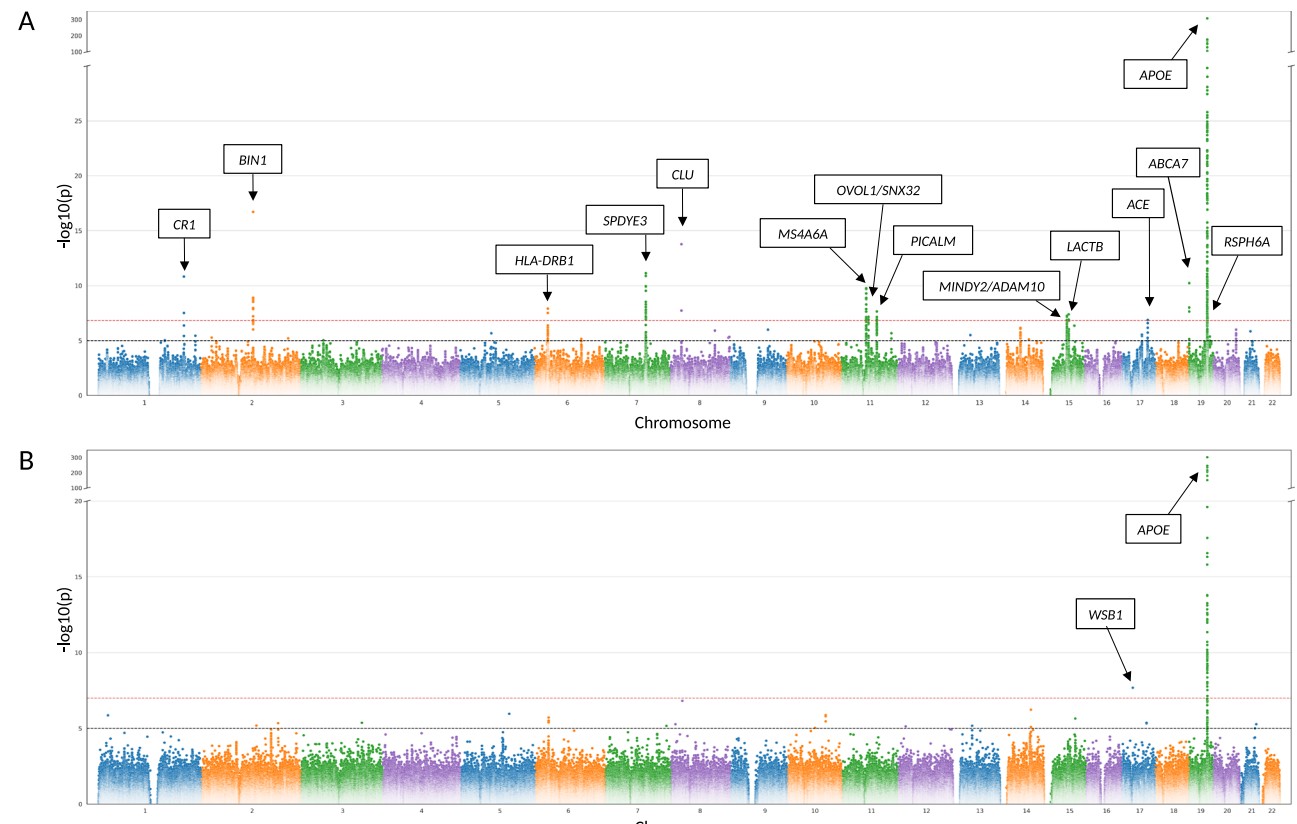

**Fig. 1 | Manhattan plots showing results of GWAS using two-sided linear regression for AD and AD-by-proxy status and imputed STRs (A) or WGS-derived STRs (B) on "White-British" cohort (based on bi-allelic STR notations).** In both GWAS, diagnosed and proxy AD cases were combined using a linear coding of the outcome (see the "Methods" section). In each box, the nearest protein-coding gene according to GENCODE V47 is annotated. Horizontal red dashed line indicates the genome-wide significance threshold (adjusted for multiple comparisons using Bonferroni correction) of 1.49E−07 (**A**) or 1.01E−07 (**B**), whereas the black dashed line indicates the suggestive significance threshold of 1.00E−05. Note that the *y*-axis is discontinuous and capped at 300. See Table 1 and Supplementary Data 2 and 12 for more details on these results. Source data are provided as a Source Data file.

Supplementary Data 5). Lastly, one variant only suggestively significant in the bi-allelic approach showed genome-wide significance in the multi-allelic approach (14:52717330). In summary, all of the 14 loci initially eliciting genome-wide significant associations in the bi-allelic screening were validated in the targeted multi-allelic analyses, either directly (same STRs, *n* = 12) or indirectly (neighboring STRs, *n* = 2), lending further support to the validity of the main findings of this arm of our study.

**Genome-wide association analyses on WGS-derived STRs**

To compare GWAS results from imputed vs. WGS-derived STR data, we utilized STR genotypes generated in the study by Halldorsson et al.[23]. After rigorous sample- and variant QC steps (see the "Methods" section), a total of 95,201 UKB participants – representing a subset fully overlapping with those in the analyses of imputed STRs (see the "Methods" section) – of self-reported "White-British" ancestry (henceforth referred to as "Halldorsson-White-British") were available for GWAS (incl. 928 AD and 13,759 AD-by-proxy cases; Supplementary Table 2). On the variant level, a total of 1,205,675 WGS-derived STRs were available for analysis. Unlike imputed STRs, for the WGS-derived STRs, both multi-allelic and bi-allelic association analyses were feasible on a genome-wide scale (see the "Methods" section). As expected based on the much smaller sample size, the number of signals was substantially reduced compared to the GWAS on imputed STRs (Fig. 1B, Supplementary Fig. 5, Supplementary Data 12). In the bi-allelic approach, 72 signals passed the genome-wide significance threshold (alpha = 1.01E−07), but only one of these (chr17:27264667 near *WSB1*)

was located outside the *APOE* region (rs429358 ± 0.5 Mb). In addition, a total of 60 WGS-derived STRs across 19 loci showed suggestive evidence of association (25 STRs at 18 loci outside *APOE*). Similarly, in the multi-allelic analyses, 38 STRs, all in the *APOE* region, passed the genome-wide significance threshold. The bi-allelic signal near *WSB1* dropped to $P = 0.55$ in the multi-allelic analyses, suggesting that for this STR the effect is driven by the size of a single allele (chr17:27264667:GA$_{16.5}$) rather than repeat length variation.

Next, we assessed why the novel genome-wide significant signal near *WSB1* did not also emerge in the GWAS using imputed STRs. Inspection of the imputed genotype data at this STR revealed that the allele (chr17:27264667:GA$_{16.5}$) giving rise to the association signal did not pass QC in the imputed STR data (due to an excess of missing genotype data). LD mapping (±500 kb) revealed no other SNPs or STRs in strong LD ($r^2 > 0.2$) with the lead variant.

Lastly, we investigated how the most promising signals near *ABCA7* and *SNX32* performed in analyses using WGS-derived STRs. Since the lead imputed STRs were not available among the WGS-derived STRs, we performed pairwise LD mapping in the ±500 kb interval surrounding these STR variants. For our signal near *ABCA7*, we identified STR chr19:1037970:T$_{14}$ > T$_{11}$ as a reasonable proxy ($r^2 = 0.65$). In the WGS-GWAS, this variant showed nearly genome-wide suggestive evidence of association (bi-allelic approach: $P = 7.68E-05$, Beta = 0.0114; multi-allelic approach: $P = 7.08E-04$), supporting the overall notion that the AD association signal near *ABCA7* is largely STR-driven. For the signal close to *SNX32*, the STR chr11:65806890:A$_{22}$:A$_{23}$ is in the strongest LD ($r^2 = 0.91$). This variant also showed strong association

**Table 1 | Summary of genome-wide significant GWAS results (bi-allelic approach) on imputed STRs in the cohort of self-reported "White-British" individuals from UKB**

| Chr. | Position (GRCh38 assembly) | STR ID | Reference allele, repeat units | Alternative allele, repeat units | Frequency (Allele 1) UKB dataset | Allele 1 | Beta | SE | P value | Gene name[a] |
|---|---|---|---|---|---|---|---|---|---|---|
| 1 | 207598421 | chr1:207598421:C:CT | $T_{12}$ | $T_{13}$ | 0.1834 | $T_{12}$ | 0.0104 | 0.0015 | 1.50E-11 | CR1 |
| 2 | 127133887 | chr2:127133887:T:TGGG | $G_5$ | $G_8$ | 0.2845 | $G_8$ | 0.0112 | 0.0013 | 1.94E-17 | BIN1 |
| 6 | 32611487 | chr6:32611487:TCTTTCTTTC:T | $(CTTT)_6$ | $(CTTT)_{3.8}$ | 0.2451 | $(CTTT)_{3.8}$ | -0.0090 | 0.0016 | 1.23E-08 | HLA-DRB1 |
| 7 | 100326718 | chr7:100326718:CT:C | $T_{19}$ | $T_{18}$ | 0.3037 | $T_{18}$ | -0.0089 | 0.0013 | 7.32E-12 | SPDYE3 |
| 8 | 27609395 | chr8:27609395:G:GA | $A_{11}$ | $A_{12}$ | 0.4021 | $A_{11}$ | -0.0093 | 0.0012 | 1.69E-14 | CLU |
| 11 | 60177578 | chr11:60177578:G:GT | $T_7$ | $T_8$ | 0.4054 | $T_8$ | -0.0078 | 0.0012 | 1.75E-10 | MS4A6A |
| 11 | 65810443 | chr11:65810443:TCA:T | $(CA)_5$ | $(CA)_4$ | 0.3849 | $(CA)_4$ | -0.0066 | 0.0012 | 7.01E-08 | OVOL1/SNX32 |
| 11 | 86092842 | chr11:86092842:TA:T | $A_9$ | $A_8$ | 0.3423 | $A_9$ | -0.0070 | 0.0013 | 2.25E-08 | PICALM |
| 15 | 58846596 | chr15:58846596:A20:A19 | $A_{20}$ | $A_{19}$ | 0.0455 | $A_{19}$ | -0.0155 | 0.0029 | 5.64E-08 | MINDY2/ADAM10 |
| 15 | 63139192 | chr15:63139192:TCC(A)22:TAC(A)19 | $A_{22}$ | $A_{19}$ | 0.4175 | $A_{19}$ | 0.0066 | 0.0012 | 4.11E-08 | LACTB |
| 17 | 63485772 | chr17:63485772:C:CA | $A_{20}$ | $A_{21}$ | 0.4550 | $A_{21}$ | 0.0063 | 0.0012 | 1.32E-07 | ACE |
| 19 | 1046580[b] | chr19:1046580:TG:T | $G_{10}$ | $G_9$ | 0.2089 | $G_9$ | 0.0096 | 0.0015 | 5.90E-11 | ABCA7 |
| 19 | 44909968 | chr19:44909968:G11:G9 | $G_{11}$ | $G_9$ | 0.1534 | $G_9$ | 0.1226 | 0.0016 | 7.65E-1201 | APOE |
| 19 | 45802824 | chr19:45802824:CTT:C | $T_{12}$ | $T_{10}$ | 0.0389 | $T_{10}$ | 0.0165 | 0.0031 | 9.49E-08 | RSPH6A |

Allele 1 – the effect allele. Beta and SE are the beta value and the standard error of the estimate in a regression t-test.

[a]Nearest protein-coding gene according to GENCODE V47.

[b]Loci that were nominated as signal-driving by COJO.

with AD in both the bi-allelic and multi-allelic analyses (bi-allelic: $P = 4.98E-04$, Beta = -0.0075; multi-allelic: $P = 2.37E-04$), albeit at reduced significance likely owing to the smaller sample size in the WGS-based analyses. The results at both loci support the notion that STRs in these regions may influence AD risk.

## Independent replication of primary STR-based GWAS results

To assess the validity of the findings in the discovery dataset, we generated analogous test statistics for all genome-wide significant STRs in an independent subset of white UKB individuals ("Other-White"; $n = 20{,}840$; 185 AD cases, 2803 AD-by-proxy cases, and 17,852 controls; Supplementary Table 1). These analyses revealed that two out of the originally identified 14 significant STRs showed direct evidence for replication when correcting for multiple testing (i.e., same effect direction and $P < 0.05/14$; Supplementary Data 6). Of the remaining 12 STRs, four show at least nominal evidence of replication ($P < 0.05$), and all but one showed the same direction of effect. Furthermore, meta-analyses across the discovery and "Other-White" subsets showed stronger associations (by $P$ value) for 10 out of these 11 loci (Supplementary Data 6). Thus, 12 out of 14 identified genome-wide significant STR loci (i.e., all but MS4A6A and SNX32) showed either direct or indirect evidence for replication in "white" individuals independent of the discovery dataset. In addition, there was also support for many STR loci in GWAS results from the other ancestry groups (incl. SNX32; Supplementary Data 6). However, we note that the sample sizes from these non-white subsets were collectively small, with AD numbers ranging from 93 to 422 (Supplementary Table 1), so replication evidence cannot be reliably interpreted in these data. Notwithstanding, the independent assessments in the "Other-White" subset of UKB revealed that most of the STR effects uncovered in the discovery GWAS analyses are stable and reproducible.

In contrast, the WGS-derived STR near WSB1 did not show evidence for association with AD risk in the "Other White" subset of the Halldorsson data ("Halldorsson-Other White"; Supplementary Table 2). However, we note that the number of AD cases in this UKB subset is exceedingly small ($n = 48$ AD cases and $n = 887$ AD-by-proxy). This, combined with the fact that the frequency of the associated allele is relatively low (MAF = 0.0145), substantially reduced the power of the replication analyses so that the lack of replication at this locus may also be a false-negative result.

## Discerning the drivers of STR-based GWAS signals

Given that the STR imputation protocol used here relies on SNP genotypes, it can be expected that some of the imputed STR GWAS signals are actually driven by SNPs at the same locus. To assess this hypothesis, we performed a range of analyses aimed at discerning SNP from STR effects (see the "Methods" section). As can be seen on the Manhattan plots from both analyses, the vast majority of genome-wide significant association peaks overlapped within a ± 250 kb window (Supplementary Fig. 6). However, fine-mapping using GCTA-COJO and SuSiE only nominated the STR in ABCA7 (COJO: $P = 5.89E-11$; SuSIE: $P = 5.90E-11$, posterior inclusion probability = 0.96) as the driver of the association signal in this region (Fig. 2), and revealed close calls for other loci, including SNX32 (Supplementary Data 7).

Next, we performed a range of analyses using two conditioning paradigms: (i) adjusting any of the 14 potential STR signals for effects of SNPs, previously identified in other studies (Bellenguez et al.[3], Jansen et al.[24]), and ii) vice versa (see the "Methods" section and Supplementary Data 8). While in the first paradigm the original STR signal dropped above the suggestive threshold (alpha = 1.0E-05) in all instances (Supplementary Data 8, Fig. 2, Supplementary Fig. 7-19), we note that two STRs (near MINDY2/ADAM10 and near ABCA7) showed a residual, almost-suggestive association after conditioning ($P = 3.36E-05$ and $P = 4.23E-05$, respectively; Supplementary Data 8, Fig. 2). Effect sizes for the STR signals slightly drop from -1.6E-02 (SE = 2.9E

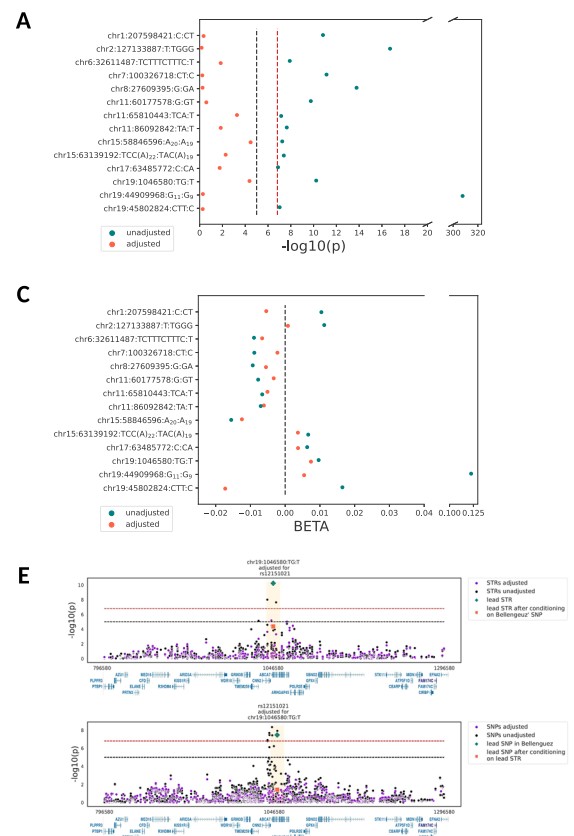

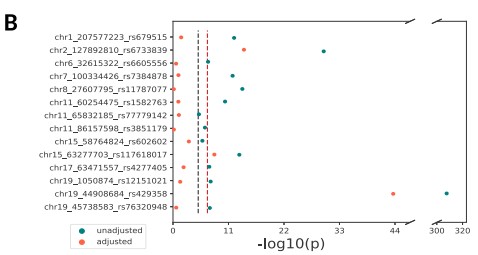

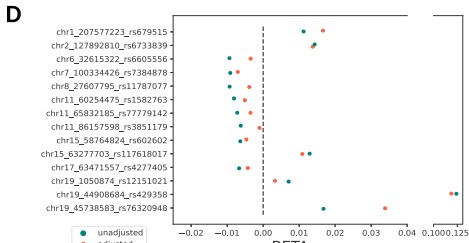

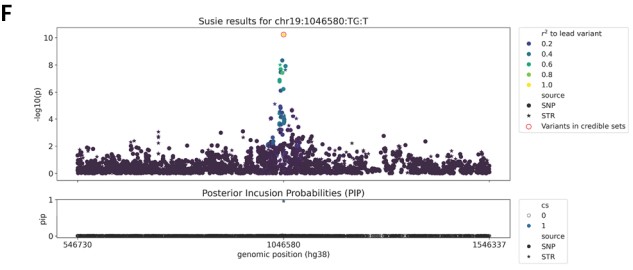

**Fig. 2 | Conditioning analyses and fine mapping results.** Summary of results performing conditional analysis using two-sided linear regression of **A**, **C** genome-wide significant imputed STRs (top hit per locus) with their corresponding SNPs, and **B**, **D** reciprocal conditioning of SNPs with STRs (see the "Methods" section). **A** and **B** show changes in $-\log 10(P)$-values, while **C** and **D** show changes in beta values. **E**, **F** Conditioning and fine mapping results for the *ABCA7* locus (chr19:1046580:TG:T). **E** The lead STR-signal was adjusted for the corresponding SNPs at the given locus (upper panel). SNPs utilized for adjustment were the lead SNPs from the largest AD GWAS published to date (Bellenguez et al.[3]). Lower panel

shows the results of reciprocal conditioning of SNP-signals with relevant STR genotypes. **F** Results of fine mapping analysis using SuSiE-RSS. STR at chr19:1046580 has the highest posterior inclusion probability (PIP) (PIP > 0.9), indicating that this is the signal-driving variant at this locus. The vertical dashed lines indicate the genome-wide significance threshold of ($P < 1.49E{-}07$; red; adjusted for multiple comparisons using Bonferroni correction) and the suggestive threshold ($P < 1.00E{-}05$; black). Blue dots: $-\log 10(P)$ values before adjustment, orange dots: $-\log 10(P)$ values after adjustment. Source data are provided as a Source Data file.

−03) to −1.3E−02 (SE = 3.0E−03) and 9.6E−03 (SE = 1.5E−03) to 7.4E−03 (SE = 1.8E−03), respectively. Residual *P* values combined with a comparatively small reduction in effect sizes indicate that STRs may substantially contribute to the known genetic association signals in these regions after accounting for the local SNP effects. Interestingly, we also observed the opposite, i.e., that certain STRs showed *stronger* evidence for association after conditioning on the effects of SNPs. This was the case for the *HLA-DRB1* (six STRs) and *ABCA7* (one STR) loci (Supplementary Data 9). A similar situation was encountered in *APOE*, where one STR (chr19:44910674) showed a change in *P* value from merely nominally significant ($P = 9.33E{-}03$) to genome-wide significant ($P = 4.67E{-}23$) (Supplementary Data 10) after conditioning on rs429358 ("e4-allele" in *APOE*). This increase in statistical support was at least partially due to the effect of the "e2-allele" in *APOE* (rs7412), which is in pronounced LD with the associated STR allele ($r^2 = 0.71$).

Interestingly, the situation of attenuated association evidence looked similar when applying the reciprocal paradigm (Supplementary Data 8, Supplementary Figs. 7–19, Fig. 2), i.e., adjusting SNP effects with STRs. Except for *APOE* and two additional SNPs, i.e., rs6733839 within the *BIN1* region on chromosome 2 and rs117618017 within the *APH1B* region on chromosome 15, all SNP signals dropped, but were still showing suggestive association signals after adjusting for STR effects, possibly indicating that STRs also contribute to leading SNP-signals. We note that a clear interpretation of these analyses is

aggravated by the fact that, by design, SNP and STR genotypes are highly correlated in this arm of our study (in contrast to analyses using WGS-derived STRs, see below). For suggestive STRs, these analyses resulted largely in a similar picture with a tendency towards more STRs "surviving" the SNP adjustment than for the genome-wide significant loci (Supplementary Data 11).

Conditional analyses on the WGS-derived bi-allelic STR signal near *WSB1* including the strongest SNP in the region (rs117222268 from ref. [3]) revealed that the STR association remained genome-wide significant after adjustment ($P = 1.77E{-}08$; Supplementary Data 13). Furthermore, of the 71 genome-wide significant WGS-derived STR signals located in the *APOE* region, we highlight STR chr19:44921083:A$_{11}$:A$_{13}$, which maps immediately adjacent to another STR (chr19:44921097-44921125:TTTA$_n$) that was already reported to be associated with AD in previous work[19,25]. Conditional analyses using rs429358 led to a drop in the STR signal to $P = 0.01$, suggesting that this STR does not elicit genetic effects over and above the well-known e4-allele. However, adjustment of the STR signals in this general region by *APOE* rs429358 led to a stark increase in effect size and association of the same STR for which a similar situation was observed in the imputed dataset (chr19:44910661, which may at least partially be due to the effect of the "e2-allele" in *APOE*, see above). Here, the *P* value increased from 1.34E−01 before adjustment to 2.38E−08 after adjustment (Supplementary Data 19). These findings, together with existing data in the

literature[26–28], suggest that the genetic architecture contributing to AD risk in the general *APOE* region may go beyond the effects of the e4-allele and deserves detailed fine mapping in future work.

## Heritability estimates for STRs compared to SNPs

To estimate the contribution of STRs to the genetic architecture of AD beyond loci showing genome-wide significant evidence of association, we computed genome-wide heritability estimates using the GCTA-LDMS method for the full dataset (see the "Methods" section). Using a population prevalence for AD of 0.05[29] and a MAF > 0.01, this resulted in an average SNP-based heritability of 37.05% (SD = 2.7%), which is in good agreement with recent estimates (31%) from other studies using GCTA for heritability estimation[30,31]. We note that the GCTA-LDMS algorithm, in general, returns numerically higher heritability estimates than the LDSC method, which is also often used in SNP-based GWAS (reviewed in ref. 30). Equivalent average heritability computations on STR genotypes yielded an estimate of 30.78% (SD = 2.8%). The average estimate, including both SNP and STRs combined, maximized at 38.09% (SD = 2.99%). Thus, STRs contribute minimally ~3% of the total AD heritability (i.e., $(h(SNP + STR)−h(SNP))/h(SNP + STR)$) (Supplementary Table 3), an estimate slightly smaller than for other complex traits[11,18]. We note that the standard deviation of the heritability estimates across all 17 batches is 2.98%, and therefore comparatively large relative to the mean. One possible reason for the high variance lies in the small relative contribution made by the STRs to the absolute heritability. As a result, the contribution metric becomes sensitive to minor natural variations between the different subsets and should, therefore, be interpreted with some caution. Notably, the estimated contribution of STRs to total AD heritability remained stable at ~3% even when alternative prevalence values were used (here: 0.1, 0.2, 0.3). Notwithstanding, given the extensive correlation structure between SNPs and STRs in this dataset, it appears possible that some genuine STR effects are masked by SNP genotypes. Hence, we consider ~3% to represent the lower bound of genetic AD heritability contributed by STRs.

## Comparison of imputed vs. WGS-derived STRs

In an attempt to assess the general validity of imputed STRs, we compared STR-based GWAS results to those using WGS-derived genotype data. To this end, we decided to primarily compare the GWAS test statistics, as a direct comparison of STR genotypes on a genome-wide level is aggravated by a number of factors (see below and the "Methods" section). As expected, the overall association evidence was less pronounced when compared to the discovery GWAS on imputed STRs (compare Fig. 1A, B), likely due to the much-reduced sample size of the WGS-derived vs. imputed dataset ($n = 95,201$ vs. 295,551, respectively). Next, we repeated the GWAS analyses on imputed STRs in the WGS subset ($n = 95,201$; Supplementary Fig. 20) and assessed the correlation of effect sizes across both sets of analyses. This comparison was limited to ~622K STRs, i.e., all those which could be uniquely matched by their genomic position and allele length (see the "Methods" section, Supplementary Fig. 22A) and revealed a highly significant correlation of effect sizes (Spearman's rank correlation) between both types of variants ($r = 0.85$, $P < 2.2E−308$). To specifically assess the validity of GWAS results for top-associated STRs, we manually curated (by matching alleles, lining up positions with UCSC coordinates, checking for SNPs possibly mapping within repeat sequences) all WGS-derived and imputed STRs with $P < 1.0E−05$ in the respective GWAS (see the "Methods" section). This resulted in $n = 84$ matching variants for which GWAS-derived effect size estimates showed a nearly perfect correlation (Spearman's $r = 0.99$, $P = 8.77E−79$; Supplementary Figs. 21 and 22B). These comparisons emphasize the overall high quality of the STR imputation process and indicate that, at least for the set of strongly associated STRs, the resulting GWAS test statistics are highly reliable.

To more directly compare the genotypes of imputed vs. WGS-derived STRs, we also compared the reference allele frequencies for both STR subsets (i.e., for the ~622K STRs matched by position and the 84 STRs matched after manual curation). Again, these analyses revealed very high correlations between imputed and WGS-derived variants ($r = 0.96$ [$P < 2.2E−308$] and $r = 0.98$ [$P = 1.12E−59$], respectively, Supplementary Fig. 23). Lastly, we evaluated imputation quality by comparing sum of allele lengths per locus between imputed genotypes and WGS-derived genotypes (Supplementary Fig. 24). Spearman correlations of sum of allele lengths per locus show a mean $r = 0.89$ and mean $r = 0.91$ for the sets of $n$ ~ 622k and $n = 84$, respectively. To enable direct comparison with data published in ref. 18, we also calculated Pearson's correlation coefficients. For the set of $n$ ~ 622k, $r^2$-values are >0.8 for 69% of STR-loci and >0.9 for 48%. While these numbers are lower than in equivalent metrics published in Margoliash et al. ($r^2 > 0.9$ for 78.7% and $r^2 > 0.8$ for 92.7%)[18], we note that our analyses are based on a substantially larger pool of variants, i.e., >600K STRs vs. only 408 in Margoliash et al.[18].

## Delineating potential functional mechanisms of AD-associated STRs

As stated above, most of the STRs identified to be associated with AD risk are located within or very close to ORFs of protein-coding sequences. Hence, a potential functional consequence of the associations may be an effect on gene expression. As a first step to assess this hypothesis, we scrutinized all STRs showing genome-wide significance in the main GWAS analyses regarding their potential to influence either DNA methylation (DNAm) or gene expression in post-mortem human brain samples (entorhinal cortex; $n = 145$ and 173[32,33]).

To this end, we analyzed all CpGs within a ±1 Mb window (i.e., those in *cis*) from genome-wide significant STR loci. After analysis, the resulting $P$ values were FDR-controlled using the Benjamini–Hochberg procedure (to 5%)[34]. Using this threshold, 6 STR loci showed associations with 31 CpG sites in the brain (of which 26 were unique; Supplementary Data 14). One of the most interesting STR meQTL findings (summarized in Supplementary Data 14) was observed with CpGs on chromosome 7q22 located between *NYAP1*, a gene that is predicted to be involved in neuron projection morphogenesis[35], and *PILRB*, a gene involved in immune system function (https://www.uniprot.org/uniprotkb/Q9UKJ0) that has already been linked to AD in several previous studies[36,37]. The meQTL association is driven by STR chr7:100326718:CT:C showing strong association with two CpGs (cg06214670 [$P = 4.63E−09$] and cg03579757 [$P = 3.27E−09$]). Furthermore, several meQTL associations were observed with CpGs mapping in or close to *HLA*-related genes (chr. 6p22), in particular *HLA-DRB1* (e.g., cg08845336 [$P = 1.33E−14$] located in exon 2 of *HLA-DRB1*), which is involved in the adaptive immune response and represents an emerging candidate gene for AD and other neurodegenerative disorders[38–40]. Lastly, we also found strong evidence for an STR-based meQTL in a proximal enhancer element of the first intron of *SNX32* (cg15531562 [$P = 7.07E−13$]), a gene involved in the regulation of neurite outgrowth[41], representing one of the novel STR-related AD loci in our discovery GWAS analyses.

In addition, we performed expression quantitative trait methylation (eQTM) analyses by probing for potential direct correlations of DNAm at all CpGs emerging from the meQTL analysis and mRNA expression for genes mapping within ±25 kb in samples drawn from the human entorhinal cortex (EC) (Supplementary Data 16). These analyses revealed two loci showing FDR-significant DNAm-mRNA correlations for several CpGs and HLA-related genes, as well as for the STR locus near *SNX32* (Supplementary Data 16). The latter was between cg15531562 (*SNX32*, intron 1) and *SNX32* transcript ENST00000308342, showing a non-canonical effect direction, where decreased methylation in parts of the gene-body is associated with lower levels of gene expression, and vice-versa[42–45]. This is precisely what we observe in our

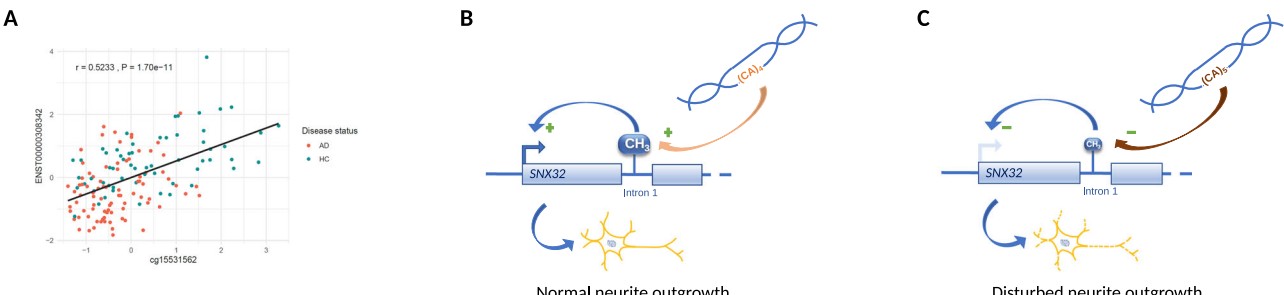

**Fig. 3 | Expression quantitative methylation (eQTM) mapping and putative mechanism underlying *SNX32* association in AD pathophysiology. A** Pearson-correlation of DNAm at cg15531562 (*SNX32*, intron 1) and the amount of *SNX32* transcript (ENST00000308342) in OPTIMA (*n* = 144) brain samples. The unadjusted linear regression line is shown for illustrative purposes. Source data are provided as a Source Data file. **B**, **C** Integrating results from GWAS, meQTL, eQTM, and eQTL analyses, we propose the following mechanism for the STR-based AD association detected for *SNX32*. The meQTL analyses suggest that the five-repeat allele at STR chr11:65810443 is associated with decreased DNAm levels at

cg15531562 (located in intron 1 of *SNX32*; beta = −0.0291, Supplementary Data 14). Further, eQTM results suggest that reduced DNAm levels lead to a reduced expression of *SNX32* mRNA (and, as a result, presumably to SNX32 protein levels). The eQTM results are corroborated by eQTL results, suggesting that the same STR that increases AD risk is associated with decreased expression of *SNX32* mRNA. This chain of associative evidence is in good agreement with the literature, which suggests that lower levels of SNX32 protein are (i) causally associated with AD[46,47] and (ii) negatively affect neurite outgrowth[41].

study, i.e., a positive correlation between DNAm and *SNX32* mRNA expression (EC: *r* = 0.523, *P* = 0.0047; Fig. 3A, Supplementary Data 16). The same CpG also showed FDR-significant correlations with transcripts in *CFL1*, although the effect size was slightly smaller than for *SNX32* (Supplementary Data 16).

Next, we computed STR-based expression QTL (eQTL) analyses using gene expression profiles derived from RNA sequencing experiments conducted in the same brain samples (*n* = 173)[33]. For each STR, we probed for associations with genes within a ±1 Mb window, again using FDR adjustment[34]. In general, there was a good correspondence between meQTL and eQTL results, and four STR-mRNA pairs attained study-wide significance after FDR correction (Supplementary Data 15). Of these, the strongest and most consistent eQTL signals were, again, observed with genes in the HLA region on chromosome 6p22. While we did not detect FDR-significant eQTL associations for *SNX32*, we note that there was a strong nominal association between STR chr11:65810443:TCA:T and *SNX32* transcript ENST00000308342 (*P* = 0.00067 with a positive effect direction [beta = 0.25], in agreement with the meQTL results). Future work in larger brain-based datasets needs to follow up on these meQTL, eQTM, and eQTL leads. As stated in the "Methods" section, we did not test combinations using other numbers of PCs in the eQTL/meQTL analysis model, which represents a potential limitation of this arm of our study.

## Discussion

In this study, we conducted an STR-based GWAS for AD in a large collection of ~330,000 UKB individuals. In analyses using imputed STR data, we identified 14 loci showing evidence for genome-wide significant association with AD risk, the strongest effects elicited by STRs in the *APOE* region. Replication testing in a subset of ~20K UKB individuals provided independent support for all but one of these 14 loci. Among all genome-wide significant signals in the UKB cohort, there was one where STRs (and not SNPs) represent the lead signal (*ABCA7*), and four where STRs show noteworthy contributions to the SNP-driven associations with AD risk (*HLA-DRB1*, *ADAM10*, *SNX32*, and *APOE*). GWAS analyses using WGS-derived STRs led to the identification of one additional genome-wide significant locus (near *WSB1*) previously not highlighted by SNP-based GWAS. Post-GWAS analyses suggest that STRs make up at least 3% of the overall genetic heritability of AD in this dataset. Lastly, aligning our top STRs in genome-wide DNAm and transcriptome profiles from independent human brain samples suggested that several of the identified STRs may unfold their effects by impacting the expression of nearby genes.

Our study reports three main noteworthy findings. First, we detected two new potential STR-based GWAS signals for AD, i.e., near *OVOL1/SNX32* on chr. 11q13, and near *WSB1* on chr. 17q11. None of the known AD SNP-based GWAS results[3] mapped within ±0.5 Mb from these loci, qualifying them as novel signals. Despite the fact that both of these loci contain promising AD candidate genes (see below), we note that neither was independently replicated in the "Other White" subset of the UKB dataset (but there was support from other ancestry groups for *OVOL1/SNX32*). The first signal was implied by an imputed STR (chr11:65810443:TCA:T; *P* = 7.01E−08) near *OVOL1* (-13.2 kb upstream) and *SNX32* (-23.5 kb downstream). While our literature search retrieved no obvious link to AD for *OVOL1*, the situation is different for *SNX32*. This gene encodes sorting nexin 32, which is very highly (and predominantly) expressed in the brain (https://www.gtexportal.org/home/gene/SNX32 and https://www.proteinatlas.org/ENSG00000172803-SNX32). The STR associated with AD also represents one of the strongest brain-based meQTL signals of this study, with CpGs annotated to *SNX32* (*P* = 7.07E−13). Furthermore, we showed that an increased level of DNAm at the STR-associated CpG cg15531562 is correlated with increased levels of *SNX32* mRNA in *post-mortem* human brain tissue (Fig. 3). Based on these results, we hypothesize that carrying the five-repeat allele at STR chr11:65810443:T(CA)$_5$ leads to decreased DNAm levels at cg15531562 (located in intron 1 of *SNX32*; beta = −0.0291), which in turn leads to a reduced expression of *SNX32* mRNA (Fig. 3). These observations fit well to two recent functional studies[46,47] showing that lower levels of SNX32 protein are causally associated with AD. Lastly, a third study[41] showed that lower levels of SNX32 protein negatively affect neurite outgrowth, again supporting our observation that increased risk at this locus may result in a lower expression of *SNX32* mRNA.

The second locus maps to chr. 17p22 and is elicited by a WGS-derived TG dinucleotide STR. It maps -29.4 kb upstream from the *WSB1* gene into an H3K27Ac mark, indicating an active enhancer element (http://genome.ucsc.edu). *WSB1* mRNA is ubiquitously expressed, including in the brain (https://www.gtexportal.org/home/gene/WSB1). Functionally, the encoded protein is a member of the WD-protein subfamily and a probable substrate-recognition component of a SCF-like ECS (Elongin-Cullin-SOCS-box protein) E3 ubiquitin ligase complex, which mediates the ubiquitination and subsequent proteasomal degradation of target proteins (www.uniprot.org/uniprotkb/Q9Y6I7). As such, it is involved in the proper functioning of the ubiquitin-proteasome system (UPS), which also plays a central role in the accumulation of pathogenic proteins in AD, including Aβ and tau[48]. Due to its low MAF (-0.01) and the complex genomic architecture in

this chromosomal region, independent replication testing was either not informative or not possible.

The second major finding of our study highlights two previously established AD genetic loci, i.e., near *ABCA7* (chr. 19p13) and *MINDY2/ADAM10* (chr. 15q21), for which STRs either represent the lead association signal or make strong contributions to known SNP effects. First, SNPs in *ABCA7* have long been established to show an association with AD risk[49]. Various comparative post-GWAS analyses suggest that STRs − perhaps more so than SNPs − may be the drivers behind this signal. A similar observation was made for the second locus, i.e., the region harboring *MINDY2* and *ADAM10*. Although the genetic evidence points more towards *MINDY2*, the second gene in this region, *ADAM10* (encoding α-secretases a disintegrin and metalloprotease 10), which functions as a constitutive α-secretase cleaving amyloid-β protein precursor (APP) protein in the non-amyloidogenic pathway[50,51], appears as the more compelling candidate.

The third main finding is related to various STR-based association signals within the *HLA/MHC* region. Here, the GWAS results on both the imputed and WGS-derived STRs suggest association signals that remain stronger after conditioning on top GWAS SNPs than vice versa, suggesting that STRs potentially make a more pronounced contribution to AD risk than SNPs. In addition, the same STRs also show strong and consistent associations with both DNAm and gene expression in human entorhinal cortex samples. All of these association signals are led by STR chr6:32611487:TCTTTCTTTC:T, which decreases risk for AD (beta = −0.0090, $P = 1.23E−08$) and DNAm (at cg26036029; beta = −0.1391, $P = 3.57E−12$) while increasing the expression of *HLA-DQA2* (transcript ENST00000374940; beta = 0.2810, $P = 2.07E−07$) in the brain. In addition, strong eQTL signals are also observed for two WGS-derived STRs in the region ($P = 1.07E−06$). Thus, there is remarkable consistency of findings in this region on the genetic (GWAS), methylomic (meQTL), and transcriptomic (eQTL) levels, strongly supporting the genuineness of this finding.

We are aware of only two other studies involving AD in STR association studies[19,20]. While Manigbas et al.[20] used a comparatively large, and partially overlapping subset of the UKB cohort (~170K individuals), their study was limited to a relatively small set (~36K) of high-quality polymorphic STRs. This may explain why the authors did not identify any genome-wide significant signals in their analyses on AD risk. In contrast, the study by Guo et al.[19] performed association testing on a comparatively large number of STRs (~300K) but only used a relatively small number of samples (~3K). Possibly because of the small sample size, Guo et al.[19] only detected genome-wide significant associations with STRs in the *APOE* region (which were comparable to our findings near *APOE*). The *APOE* STR highlighted by Guo et al.[19] (chr19:44921098), also shows genome-wide significance in our WGS-based data; indeed, with $P = 2.99E−236$ it represents the third strongest signal in the WGS-based arm. Furthermore, it is only 14 bp apart from the most significant WGS-based STR in our data (chr19:44921083, $P = 5.69E−303$). Pairwise LD analysis shows a strong correlation between variants ($r^2 = 0.75$ and $D' \sim 1$), indicating that both STRs likely capture the same association signal. By implementing STR imputation, our study maximized power by using both a larger number of samples (~330K) and STRs (~3M variants, of which 335K were independent) than the previous papers.

Despite its strengths, our study also has several potential limitations. First, our main and most powerful GWAS analyses (in ~330,000 individuals) are based on imputed STR genotypes. While comparisons between AD-associated STRs from imputation and WGS indicated a high correlation of GWAS test statistics (confirming the validity of our main findings), it is important to note that current STR imputation paradigms do not yet achieve the same high quality achieved for imputation of SNPs[16]. Second, to streamline the statistical analyses, we split multi-allelic STRs into bi-allelic STRs, which might have reduced our power to detect association signals at some loci. We note,

however, that this procedure does not invalidate any of the bi-allelic association signals reported herein. For associated STRs in the bi-allelic tests, we then computed targeted multi-allelic association analyses. These confirmed all initially identified STR loci except two, both located within the same 4.5 Mb interval on chromosome 15q21-22 (between *MINDY2/ADAM10* and *LACTB*). The non-confirmation of the bi-allelic findings on chr.15 could be due to the possibility that the bi-allelic signal represents the effect of one allele that is "fixed" in a common haplotype, while there is no effect of the other alleles at these STR loci. We note, however, that this does not invalidate the initial bi-allelic results in these regions, as we identified other highly significant multi-allelic signals with other STRs that mapped into the same loci. Third, the multi-allelic nature of the underlying STR genotype data did not allow running generalized mixed model approaches, which permit the inclusion of related individuals. As a result, our GWAS computations were restricted to unrelated individuals, which led to a reduction in the effective sample size (and statistical power) compared to standard SNP-based analyses. Fourth, the proxy phenotyping approach used was recently shown to be the source of some bias in the context of GWAS, mostly by potentially inflating or skewing estimates of genetic correlations between traits[52]. It is important to emphasize that only a negligible bias was detected for the *discovery* of GWAS loci[52], which is how proxy phenotyping was used here. This is also what we observe in our study, where effect sizes (beta values) of GWAS analyses computed with and without the ~43K proxy AD cases correlated (Pearson's $r = 0.8$ for all and $r = 0.99$ for at least suggestively significant variants). Fifth, our analyses mainly focus on UKB individuals of European ("white") ancestry. By a margin, these ancestry groups represent the largest subsets within the UKB release included here. While several of our top STR GWAS signals either directly or indirectly replicate in other ancestries available in UKB, these analyses are based on comparatively small sample sizes (containing maximally ~400 AD and AD-by-proxy cases). Sixth, for this study, we did not consider rare STRs. Although such STRs would be highly interesting, low-frequency STRs were only available in a "Halldorsson-White-British" subsample. The sample size of only 928 diagnosed and 13,759 proxy AD cases would provide very low power to detect genome-wide significant associations, which is why we decided against including them in our GWAS analyses. Lastly, while we were able to delineate some initial hypotheses on putative mechanisms underlying our top STR-based association signals by using genome-wide DNAm and transcriptome profiles in an independent set of human brain samples, this latter dataset is comparatively small ($n_{max} \sim 173$) and only includes one brain region (entorhinal cortex).

In conclusion, our study constitutes a large-scale STR-based GWAS for AD risk. In addition to detecting two potentially novel and functionally interesting AD loci (near *SNX32* and *WSB1*), our results suggest that STRs may strongly contribute to the genetic effects at four additional loci (*ABCA7*, *MINDY2/ADAM10*, *HLA-DRB1*, and *APOE*). Furthermore, we estimate that STRs account for at least 3% of the genetic variance underlying AD. Future studies, ideally using directly genotyped STRs in large AD datasets, are needed to confirm our results and to further delineate the likely sizeable role that STRs play in the genetic makeup of AD.

## Methods

### Ethics

The study design and conduct complied with all relevant regulations regarding the use of human study participants. Collection of the UKB data was approved by the Research Ethics Committee of the UKB under application ID 81874. For the Optima Dataset, the Ethics Committees of Oxford University and the University of Lübeck approved the use of the human tissues. The overall study was conducted in accordance with the Declaration of Helsinki, and participants for both the UKB and OPTIMA data gave informed consent.

## Human subjects

The UKB contains data for 502,364 samples, of which 488,377 have array SNP data available (Field ID-22418). We checked the 488,377 samples for mismatches in reported and genotyped sex, aneuploidy, and heterozygosity outliers using genomic QC-tests that are provided within the UKB (Field IDs 22001, 22019, and 22027), removing 1805 samples in total. Of the remaining 486,539 samples, 3783 are classified as AD cases based on ICD-10, while 482,539 are non-cases. Samples without an ICD-10 diagnosis of AD were further subjected to "proxy phenotyping" (see below). To this end, we check whether they provided information about parental age (Field IDs 1845 [2526] and 2946 [1807]; "mothers/fathers age [at death]") and AD status (Field IDs 20107 and 20110; "Illnesses of father/mother", AD/dementia coded as 10 [data-coding 1010]). This information was not available for 93,260 samples, which were subsequently removed from all analyses. Next, we checked all remaining samples for possible relatedness using UKB data field 22021. Participants with no relatives identified were kept ($n = 273,321$), and samples with 10 or more relatives were removed ($n = 169$) without further analyses. For the remaining 119,572 samples with at least one but <10 relatives according to UKB data field 22021, we ran kinship removal tests based on the KING algorithm provided by PLINK2 (v2..00a3LM, www.cog-genomics.org/plink/2.0/)[53], using a threshold of 0.025 (--king-cutoff 0.025). We prioritized cases and proxy-cases above controls, removing 59,447 additional samples. This procedure resulted in a final dataset of 333,446 samples after QC.

The 333,446 samples were split into groups by self-reported ethnic background (UKB field ID 21000_i0). For the participants of self-reported "White" ancestry, we split the cohort into the subset of "White-British" ($n = 295,551$; "discovery" sample in GWAS) samples and combined the remaining "White" samples (i.e., "Any other white background" and "White-Irish") into the cohort of "Other-White" ($n = 20,840$; "replication" sample) individuals. These two sub-samples are independent. Due to their sample size, other groups ("Asian", "Black", "Chinese", "Mixed", "No_Answer", and 'Other') were not further subdivided. All samples were additionally filtered according to the availability of WGS-derived STR genotypes as determined by Halldorsson et al.[23], leaving 107,289 samples for that analysis arm.

A visual summary of this workflow can be found in Supplementary Fig. 1.

## Procedure for "proxy phenotyping"

The "proxy phenotyping" approach applied here was originally developed and validated by Liu et al.[54], and subsequently applied to all major AD GWAS using UKB data[3,24,55]. UKB provides information about participants' health records in the form of ICD-10 codes. Samples with any form of AD diagnosis by ICD-10 (Code F00: 'Dementia in Alzheimer's disease'; Code G30:' Alzheimer's Disease') were considered to be "diagnosed AD cases" ($n = 3783$).

Proxy Scores were then assigned to the remainder of the dataset based on self-reported parental AD status. Each participant not categorized as "diagnosed AD case" was assigned an "AD proxy score" ranging from 0 to 2. The algorithm to calculate proxy scores was derived from Jansen et al.[24]. Proxy scores are composed of parental risk scores for AD, ranging from 0-1 per parent, where parents who have a recorded form of any AD are scored as 1. Otherwise, they get a risk score of their age relative to 100, capped at 0.32 to not exceed the maximum prevalence of AD as reported in Herbert et al.[56], such that $\text{risk}_{\text{parent}} = \min((100 - \text{age}_{\text{parent}})/100, 0.32)$ and $\text{proxy\_score} = \text{risk}_{\text{mother}} + \text{risk}_{\text{father}}$. Next, scores $\leq 0.64$ (i.e., samples without any parent affected) were considered as "controls" while scores between 0.64 and 2 (i.e., samples with at least one affected parent) were considered as "proxy cases".

To account for the certainty of the disease status, "ICD-10 diagnosed AD cases" were assigned a proxy score of 4, i.e., they carry twice

the weight as the highest-scoring proxy AD cases in the subsequent association analyses. Note that this differential weighting is not implemented in the procedure by Jansen et al.[24], who assign equal weights to proxy cases with two affected parents and cases with an ICD-10 diagnosis of AD. We strongly believe that this approach of assigning higher weights to diagnosed AD cases vs. proxy AD cases better reflects the clinical reality.

## STR data processing and quality control

**STR imputation and QC.** For STR imputation, we followed the approach published by Bustos et al. and Saini et al.[11,15]. This is based on genome-wide SNP genotyping data released by UKB (field ID = 22418) and the 1kGP SNP-STR haplotype panel released by Ziaei Jam et al.[16]. Before imputation, SNPs are filtered for genotyping efficiency (exclude all with <98%) and for minor allele frequency (exclude all with <1%), followed by conversion to vcf-format and lifting from hg19 to hg38 using GATK LiftoverVcf[57]. To reduce computation time, samples were divided into batches of $n = 5000$ before imputation and matched with the imputation panel using the conform-gt tool. STR genotype imputations were computed using Beagle V5.4[58]. After imputation, all SNP genotypes were removed, followed by splitting of multi-allelic STRs into a bi-allelic format and filtering for imputation quality (exclude all variants with $DR^2 < 0.7$). Due to imperfections in the imputation catalog, the QC'ed data still contained genotypes on a number of non-STR variants, e.g., indels and SNPs, and duplicate STR entries, requiring manual inspection and removal. To facilitate this process, non-STRs were removed after GWAS from the set of either genome-wide significant or suggestive association signals.

**QC of WGS-derived STRs.** UKB also provides WGS-derived (i.e., non-imputed) STR data for 150,000 participants published in a UKB analysis of WGS data by Halldorsson et al.[23] (Field ID 23365). Before analysis, the WGS-derived STR data were QC'ed and split into bi-allelic variants, as outlined above for imputed variants.

**QC of genome-wide SNP genotyping data.** For SNP-based GWAS analyses, imputed SNPs provided by UKB (Field ID 22828) were used. First, they were lifted from hg19 to hg38 using GATK LiftoverVcf[57] and then filtered according to the following criteria using PLINK (v2.00a3LM, www.cog-genomics.org/plink/2.0/)[53]: All samples are split into the respective cohorts, variants with minor allele frequency below 1% and genotyping efficiency below 2% are filtered (--maf 0.01 --geno 0.02). Also, variants with deviations from HWE at $P < 5E-06$ (--hwe 5e-06) in controls were removed.

## Genome-wide association analyses

Association tests were run for imputed STRs, imputed SNPs, and WGS-derived STRs in PLINK (v2.00a3LM) using linear regression models on the proxy phenotyping score (see above), adjusting for genetic ancestry (using the first 15 principal components [PCs] from a principal component analysis on LD-pruned non-imputed SNP genotype data) and sex. For the SNP-based data (i.e. imputed STRs and SNPs) we additionally adjusted for genotyping-array (field ID 22000). For the WGS-derived STR data, we adjusted for sequencing provider (200K-release; field ID 23080). To define a genome-wide significance threshold, we estimated the number of independent ($r^2 < 0.2$ or more than 1500 kb apart) imputed STRs to be $n = 334,605$ and used this number for determining a Bonferroni-corrected significant threshold of $\alpha = 1.49E-07$ ($\alpha = 0.05/334,605$). Similarly, for the WGS-derived STRs, we identified $n = 495,465$ independent STR-variants resulting in a Bonferroni-corrected significant threshold at $\alpha = 1.01E-07$ ($\alpha = 0.05/495,465$).

After GWAS for imputed STRs, since the STR reference panel used for imputation[16] contains several imperfections, we performed a manual check for all genome-wide significant and suggestive signals

based on each STR's genome coordinate. This led to the removal of 223 variants that were duplicated (with different IDs) and 370 non-STR variants (indels or SNPs), leaving 482 unique STRs (254 genome-wide significant STRs at 14 loci and 228 suggestive at 21 loci).

We also performed sex-stratified (male and female; $n$(male) = 131,342, $n$(female) = 164,209) GWAS analyses for imputed STRs to search for potential sex-specific loci. In addition, we ran a genome-wide interaction analysis using "genotype × sex" as an interaction term to search for loci that show substantial differences in the association results in males vs. females.

To assess potential biases introduced by the use of proxy phenotypes, we repeated the GWAS analyses for the imputed STR variants using only diagnosed AD cases ($n$ = 2947) and controls ($n$ = 249,681). The results from this GWAS were used to compare the effect estimates to those from the GWAS with proxy cases by computing Pearson's correlations across both datasets.

Given that age is the strongest risk factor for AD, we performed sensitivity analyses by re-running GWAS analyses with age (i.e., age at recruitment, data-field 21022; NB: the UKB catalog does not provide "age at onset" for AD as an additional variable, hence age at recruitment was used, in agreement with previous AD GWAS[24]) included as a covariate. However, we do not consider age as a classical confounder in this context, as it is not assumed to be associated with the predictor variable (genotype). To assess the effect of age on the results, we calculated Pearson correlations of beta values with and without adjustment for age and quantified a potential gain in precision (ratio = SE_age/SE_no_age).

## Replication analysis

To assess the validity of the primary GWAS findings in the discovery dataset of $n$ = 295,551 "White British" UKB individuals, we used the $n$ = 20,840 "Other-White" subset as a replication cohort. These were analyzed using the same GWAS analysis paradigms as for the discovery computations. To count as "direct replication", variants needed to show at least nominal evidence of association corrected for multiple testing ($P < 0.05/14$) and the same direction of effect as in the primary GWAS. Lastly, results from both discovery and replication datasets were combined by fixed-effect meta-analyses using PLINK (v1.90b7). In this context, we considered variants showing stronger statistical support (=smaller $P$ values) in the meta-analysis as compared to the discovery results but not fulfilling the above replication criteria as "indirect replications".

## Multi-allelic analysis

To ensure that splitting of multi-allelic variants into a bi-allelic coding does not introduce false positive findings, we repeated association analyses for all variants ± 500 kb around STRs leading to genome-wide significant or suggestively significant loci using associaTR[59]. AssociaTR performs ordinary least squares regression using the sum of allele length as genotype. Here, we performed two types of analyses for the multi-allelic variants: (i) direct association analyses, i.e., using imputed variants without multi-allelic splitting or rejoining of bi-allelic variants, (ii) two-step association analyses. While running the analyses for (i), we noticed that some variants appear in both (that is, multi- and bi-allelic) formats in the reference catalog. For this analysis arm, we therefore selected the multi-allelic variant from the catalog wherever possible (bcftools view -m3). All remaining variants were computationally "rejoined" into a multi-allelic format (bcftools norm -m+any) from the bi-allelic notation, which we used in the main GWAS analysis arm.

Association analysis using associaTR for WGS-derived STRs was performed for the whole genome. Here, variants passing QC for the bi-allelic branch (see the section "STR data processing and quality control" with subsection "QC of WGS-derived STRs") were rejoined into a multi-allelic format (*norm -N -m+any*). Association testing was run using associaTR and the same set of covariates as in the bi-allelic study.

## Fine mapping and conditional analyses

A range of analyses aimed at discerning SNP from STR effects was performed. This entailed running a SNP-based GWAS on the discovery dataset using imputed genotypes released by the UKB.

To identify lead variants within the imputed STR data, we used GCTA-COJO[60]. The same tool was used on combined SNP and STR GWAS summary statistics to distinguish STR and SNP signals in our results.

To identify lead STR variants, we determined the most significant variant within any given 250 kb window by checking whether there was any variant with a smaller $P$ value in this vicinity.

Additionally, we used sum of single effects regression (SuSiE-RSS)[61,62] for fine mapping based on summary statistics. To this end, we included both STRs and SNPs mapping within ±500 kb around the lead STR per locus. For these combined SNP and STR variants, we then calculated pairwise LD matrices and ran SuSiE-RSS. For each locus, we defined the variant with the highest posterior inclusion probability (PIP) across all models as "lead variant".

In addition, we performed two types of conditional analyses to compare SNP vs. STR effects: (i) we adjusted the STR signal for the effects of the respective lead SNP (±500 kb) per STR locus derived from the SNP-based AD GWAS by Bellenguez et al.[3]. This was done for all genome-wide significant loci except *APOE*, which was excluded in the study by Bellenguez et al.[3]. Here, we adjusted for the well-established epsilon-4 AD risk allele (i.e., T-allele at rs429358). For the STR signal at chr19:45802824 (which was also excluded in Bellenguez et al.[3] due to its proximity to *APOE*), we adjusted for rs76320948, which was identified as the lead variant by Jansen et al.[24]. (ii) We performed reciprocal analyses where the SNP association statistics in any of the top STR loci were conditioned on STR genotype. In case the top regional Bellenguez SNPs did not fulfill our variant QC-criteria (MAF < 0.01 and genotyping efficiency < 0.98) in the UKB SNP dataset, we searched for the next strongest SNP until the criteria were fulfilled.

## Heritability analysis

Heritability analyses were run using the GCTA-LDMS method[63], following the protocol provided by the authors. Here, a segment-based LD score is calculated first, based on which all variants are stratified into four LD groups. One advantage of this method is that it prevents LD bias among variants imputed from SNPs, which may occur using the regular GCTA algorithm[63]. To increase the precision of the resulting estimate and to improve computational efficiency, the heritability calculations were run in multiple batches on a subset of individuals within the discovery cohort of "White British" individuals, i.e., all $n$ = 2947 diagnosed AD cases merged with different batches of 15,000 randomly selected controls. We repeated these runs for 16 batches on non-overlapping control samples. This procedure left $n$ = 9681 controls, which were placed in a 17th batch, supplemented with randomly chosen controls from the other batches to ensure equal batch size. Heritability analyses were run for SNPs and STRs individually, as well as the combined SNP and STR data. Contribution of STRs to the total AD heritability was calculated as follows: ($h$(SNP + STR)−$h$(SNP))/ $h$(SNP + STR).

## Comparison of GWAS results for imputed vs. WGS-derived STR variants

To assess imputation quality, we computed Pearson's and Spearman's correlations for the sum of allele length per sample and calculated Spearman's correlations for allele frequencies and GWAS beta values for all matching variants on the same subset of UKB individuals ($n$ = 95,201; "Halldorsson subset"). Identifying matching STR counterparts in the imputed vs. WGS data is aggravated due to the fact that the STRs originate from different toolkits and catalogs (popSTR for WGS-derived data and HipSTR for imputations). Discrepancies in the

sequences for reference and alternative alleles result in different variant positions and IDs, which makes it difficult to match them directly. To overcome this problem, we matched the STRs data using two approaches. (i) Genome-wide: Here, variants originating from WGS and imputation were matched algorithmically by genomic position and length of reference and alternative alleles, resulting in a total of $n = 622,155$ variants that could be compared. However, variants with slight differences in genomic position are not matched using this procedure. Therefore, in approach (ii) we manually curated all STRs passing at least suggestive significance in both imputation- and WGS-based GWAS. Manual curation was performed with respect to "true" allelic matches (i.e., after accounting for SNPs mapping within the repeat sequences, correcting reference allele assignments, or correcting small shifts in genomic position). Eventually, this led to the definition of $n = 84$ manually curated, perfectly matching variants. Manual curation was performed using results of the full sample set ($n = 295,551$) for imputation-based GWAS vs. the subset of $n = 95,201$ samples for the WGS-based results. A flow chart depicting the selection and matching process is provided in Supplementary Fig. 21. The resulting, truly matching variants were also tested for correlation of beta values and sum of allele length as well as allele frequency.

## Methylation quantitative trait locus (meQTL) analysis

To investigate the potential association between STR genotypes and DNA methylation (DNAm) levels, we performed meQTL analyses targeted towards CpGs located in the immediate vicinity (i.e., ±1 Mb) of genome-wide significant STRs emerging from the main GWAS. DNAm profiles were assayed using the "Infinium MethylationEPIC" array (Illumina, Inc.) on DNA extracts from human brain tissue (entorhinal cortex, Brodmann area BA28) of 145 AD cases and controls from the Oxford Project to Investigate Memory and Aging (OPTIMA) dataset[32]. This dataset was optimal for the assessments in this project owing to the parallel availability of genome-wide SNP genotype (allowing STR imputations) and DNAm and mRNA expression data (allowing the various QTL analyses described below). STR genotypes were imputed from genome-wide SNP genotyping data (Global screening array [GSA]; Illumina, Inc) using the same procedure outlined above for the UKB samples. Details on the OPTIMA dataset, as well as the procedures used for DNAm profiling, can be found in Sommerer et al.[32]. In brief, this dataset was drawn from the larger longitudinal, prospective Oxford Project to Investigate Memory and Aging (OPTIMA). Snap-frozen post-mortem human EC slices from Brodmann area 28 (BA28) were obtained from the Oxford Brain Bank for both AD patients and cognitively normal elderly controls. Details regarding the computational packages used, QC procedures, and the general workflow of the meQTL analyses can be found in Ohlei et al.[12]. Briefly, we used the R-package "MatrixeQTL" applying linear regression models adjusting for sex, the first two principal components (PCs) of the genotype data, and the first three PCs of the DNAm data. Specifically, the meQTL results were taken from a separate project describing the results of genome-wide STR-based meQTL mapping in the OPTIMA dataset (manuscript in preparation). Our choice of how many PCs to include in the model was based on scree plots made from the quantified mRNA data. We did not test combinations with other numbers of PCs for this analysis, which represents a potential limitation of this arm of our study.

## Expression quantitative trait locus (eQTL) analysis

To investigate the potential association between STR genotypes and gene expression levels, we performed eQTL analyses targeted towards mRNAs located in the immediate vicinity (i.e., ±1 Mb) of genome-wide significant STRs emerging from the main GWAS. mRNA levels were drawn from transcriptome-wide gene expression data derived from total RNA sequencing in the same dataset of OPTIMA brain samples ($n = 173$) also used for the meQTL analyses (for details on processing and QC of these RNA sequencing data see ref. [33]). Analytical procedures for the eQTL analyses were equivalent to those described for the meQTL analyses above, i.e. we applied linear regression models including sex, the first four principal components of a PCA assessing genetic ancestry, and the first two principal components of RNA levels as covariate. Our choice of how many PCs to include in the model was based on scree plots made from the quantified mRNA data. We did not test combinations with other numbers of PCs for this analysis, which represents a potential limitation of this arm of our study.

## Expression quantitative trait methylation (eQTM) analysis

For CpGs emerging from meQTL analysis as significantly associated with STR genotypes, we additionally assessed potential correlations between DNAm and mRNA data for transcripts mapping within ±25 kb of the corresponding CpG. For these analyses, we used the same RNA seq and DNAm EPIC array data as outlined above, in OPTIMA samples where both types of data were available ($n = 144$). Analytically, these eQTM computations are based on linear regressions using DNAm (i.e., beta values) as a predictor and mRNA levels (transcripts per million) as outcome, adjusting for age, sex, AD disease status, RNA integrity number (RIN), and postmortem interval (PMI). To account for potential confounding due to technical variation (e.g., laboratory batch effects) and biological factors (e.g., cell-type composition), principal components derived from a principal component analysis of both DNAm and mRNA data were additionally included as covariates. Specifically, we included three DNAm PCs and one mRNA PC as covariates in the regression model.

## Application of the analysis code

No new software was developed for the analysis of this project. Instead, all software tools were previously published. Relevant references as well as parameters for their execution are provided in the methods section. Exemplary code used to perform the analyses described in this study is available at https://github.com/dgmelin/ukb_ad_str_gwas (https://doi.org/10.5281/zenodo.19484860[64]).

## Reporting summary

Further information on research design is available in the Nature Portfolio Reporting Summary linked to this article.

# Data availability

Individual-level genetic and phenotypic data used in this study are available under restricted access due to the legal terms of the UK Biobank Material Transfer Agreement from the UK Biobank (https://www.ukbiobank.ac.uk/enable-your-research/register). Access to the individual-level genetic data from all participants can be obtained by qualified researchers by applying directly to the UK Biobank. The individual-level imputed STR data generated in this study will be returned to the UK Biobank. Access can be obtained by approved researchers through the UK Biobank as a 'Returned Dataset' (under application ID 81874) once UKB's internal processing is complete. The raw individual-level genetic data are protected and are not publicly available due to data privacy laws. The SNP-STR imputation panel is available in the Zenodo database under accession code https://doi.org/10.5281/zenodo.8365671. Access to the OPTIMA dataset is restricted due to data protection regulations. Principal investigators can be contacted with data access requests via their research group portal (https://www.pharm.ox.ac.uk/research/groups/smith-group-oxford-project-to-investigate-memory-and-ageing-optima-and-b-vitamin-research-group), subject to the execution of a standard data use agreement. GWAS summary statistics generated in this study have been deposited in the Zenodo database under accession code https://doi.org/10.5281/zenodo.17908176. Source data for the main figures are provided with this paper, while source data for the Supplementary

figures are included in the Zenodo repository. The previously published GWAS summary statistics from Bellenguez et al.[3] used in this study are available in the GWAS Catalog database under accession code GCST90027158 http://www.ebi.ac.uk/gwas/publications/35379992. Summary statistics from Jansen et al.[24] have been retrieved from https://doi.org/10.1038/s41588-018-0311-9. Source data are provided with this paper.

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

## Acknowledgements

This research has been conducted using the UK Biobank Resource under Application Number 81874. We acknowledge the Oxford Brain Bank, supported by Brains for Dementia Research (BDR) (Alzheimer Society and Alzheimer Research UK) and the NIHR Oxford Biomedical Research Center. The authors thank Dr. M. Volkan Cakir and Ms. Ida Happel, who supported some of the analyses performed in this study, and Prof. Bernhard Haubold for helpful discussions.

## Author contributions

Overall study design: V.D. and L.B. Data acquisition and processing: D.G., D.P., K.M., M.P.J., L.P., C.L., R.E.T., V.D., L.B. Statistical analyses: D.G., O.O., M.M.A., M.P.J., V.D. Interpretation of findings: D.G., R.E.T., V.D., L.B. Writing of first draft: D.G., V.D., L.B. Review, editing, and approval of final draft: all authors.

## Funding

L.B. and V.D. disclose support for the research of this work from the Deutsche Forschungsgemeinschaft [DO2354/1-1 to V.D. and BE2287/9-1 to L.B.] and the Cure Alzheimer's Fund as part of the Alzheimers Genome Project™. C.M.L discloses support for publication of this work from the Heisenberg Program of the DFG [LI 2654/4–1.9]. Other authors declare no relevant funding. Open Access funding enabled and organized by Projekt DEAL.

## Competing interests

The authors declare no competing interests.
