## [Transparent Peer Review file · Nature Communications]

GWAS on short tandem repeats identifies genetic mechanisms in Alzheimer's disease

Corresponding Author: Professor Lars Bertram

Version 0:

Reviewer comments:

Reviewer #1

(Remarks to the Author)

In the paper entitled "GWAS on short tandem repeats identifies novel genetic mechanisms in Alzheimer's disease", the authors describe results of genome-wide associations of biallelic STRs with AD, for sample sizes reaching up to 330,000 individuals from the UK biobank. Tests were performed using imputation-derived STRs and, separately, WGS-derived STRs. As a result, authors identified 14 genome-wide significant hits using imputed calls (higher sample sizes) and 1 hit using WGS-derived calls. The authors also tested for replications in an independent set of individuals, showed the additional effects of STR to AD heritability, and explored potential functional implications of their findings via eQTL and meQTL mapping.

Results regarding the implication of STRs in AD are relevant and much appreciated by the field. The methodology applied in this study was adequate. However, the overall presentation is subpar to what readers might expect in Nat. Communications. Many possibly interesting results are buried in supplementary materials, and the description of results can be improved to better highlight the strengths and findings.

Below are some points I believe can improve the current stage of the manuscript.

The assumptions of findings resulting from imputed- and WGS-derived STRs are quite distinct. GWAS of imputed-STRs will inform if known GWAS SNP hits are in LD with STRs, which will be informative for fine-mapping. Although the conditional associations used in the manuscript are relevant, I believe that applying functional fine-mapping strategies has a better chance of capturing the real causal variants.

Regarding the WGS-derived STRs, we can expect to find novel signals (with appropriate caution), considering that rare STRs are better captured and variants in more complex genomic regions are usually not well mapped in SNP arrays. Although the authors somewhat discuss these differences, I feel this should be more evident in the way the results are organized and presented. Currently, it looks as if the same analysis was just performed twice with two different sets of variants.

One important technical choice taken in the current paper was to split the repeats into biallelic variants, rather than to consider the actual lengths. Currently, it feels like this decision was made purely by convenience. There are obvious differences in considering either option, and both strategies could be complementary. For instance, by considering repeat lengths, you might capture effects of repeat expansions, while just biallelic variants probably represent repeats "fixed" in common haplotypes.

The present study is very closely related to two papers referenced by the authors (both published in Nat. Communications). PMC11614882: "A phenome-wide association study of tandem repeat variation in 168,554 individuals from the UK Biobank"; and PMC11775329: "Polygenic burden of short tandem repeat expansions promotes risk for Alzheimer's disease". Both studies considered repeat lengths rather than genotyping alleles, besides other differences. The authors should make an effort to at least compare findings with previous related studies. This not only helps to understand the strengths and weaknesses of each approach but also strengthens the evidence for their main findings.

The results section starts by describing the results of the first GWAS. Starting with a summary describing how the paper will

flow might help readability. If possible, please provide some general information about the STRs identified before jumping into association tests. Again, make a clear separation of the purposes of each section. Please bring more details of the main findings to the main Figures. The current paper has only 2 main figures (2 Manhattan plots) that are ok to show, but they are basically illustrative.

The current interpretation of the results is quite simplistic. There are many locus zoom plots in the supplementary material that could help, but they still need visualization improvements. Also, for comparison, plotting results of an independent AD GWAS of the same locus (aligned with STR results) will help to highlight cases where the signal was replicated.

Here are some more focused points that the authors could consider.

Genome-wide association analyses on imputed STRs

Line 110: If this STR association is real, we should expect to observe SNP associations in the locus. I suggest checking the AD GWAS complete summary stats.

Discerning the drivers of STR-based GWAS signals

Line 148: The ABCA7 result is possibly interesting; more details could be shown in the main figures.

Heritability estimates for STRs compared to SNPs

Line 181: The authors found that heritability based on SNP resulted in ~36%, while using only STRs resulted in ~31%. This is quite a high estimate, which is probably explained by APOE alone. This should be acknowledged.

Genome-wide association analyses on WGS-derived STRs

Line 196: Unclear if "Halldorsson-White-British" is a subset of "White-British" or a totally different individuals.

Line 199: Were less frequent STRs also considered? This is an advantage of using WGS-derived variants.

Comparison of imputed vs. WGS-derived STRs

Line 249: Why only 84 variants matched? It would be interesting to show the concordance of allele frequencies between strategies.

Delineating potential functional mechanisms of AD-associated STRs

Line 259: Not clear if DNAm and RNAseq were performed in the same individuals used in the GWAS. If it is an independent dataset, why was this particular one chosen? There are other sources of omics data with a better sample size that could be used for this purpose.

Methods

Line 518: What threshold of kinship was used?

Line 538: This strategy was derived by Jansen et al. Do other AD GWAS apply similar strategies?

Line 560: By splitting multiallelic STRs into biallelic variants, the information regarding potential repeat expansions is lost. So, basically, the STRs surviving QC are "fixed" insertions of repeat patterns. Please include some considerations regarding these underlying conditions.

Line 636: Randomly subsetting of individuals raises questions about the robustness of the heritability estimates. Please show that this is not biased.

Line 645: Although it's not a straightforward task to compare different STR calls, using different IDs as an explanation is not adequate.

Line 648: Given the simplistic condition used to match variants, please provide a comparison of allele frequencies of matched STRs before and after manual curation.

Line 653: What is meant by targeted meQTL analysis?

Lines 660 and 662: The DNAm data analyzed was generated by Sommerer et al., while the QTL methodology followed Ohlei et al.? If that is the case, what is the ms in preparation (line 665) about? The text is not clear on what was done in the present study vs. what was obtained.

Line 664: Were other numbers of DNAm PCs tested? Usually, this improves meQTL detection.

Line 668: What is meant by targeted eQTL analysis?

Line 671: I understand this is re-using data, but a brief description is welcome.

Line 674: Were other numbers of RNA PCs tested? Usually, this improves eQTL detection.

Line 676: Please make summary statistics publicly available.

Line 683: Please make the code used for analysis publicly available.

Reviewer #2

(Remarks to the Author)

In this manuscript, Gmelin and colleagues perform genome-wide associations between STRs and Alzheimer's disease risk in the UK Biobank. Their work assessed a large number of STRs genome-wide using both imputed STR genotypes and WGS-derived genotypes. They identify a number of STR disease associations, many of which are at known AD GWAS loci. At several loci, the STR associations appear to be at least partially conditionally independent from known SNP associations, suggesting that the STRs may have a causal role at the loci. Their work extends upon recently published findings linking STRs with AD risk. However, there are a number of major issues that need to be addressed:

- The use of a biobank results in an unbalanced case/control ratio which can lead to skewed test statistics. Saddlepoint approximation and other methods have been used in these scenarios to reduce Type I error. The authors should demonstrate that their associations are not the result of a highly unbalanced case/control ratio.
- Recent work has suggested potential biases in GWAS by proxy for Alzheimer's disease (see PMID: 39496879). The authors should demonstrate that their results are not spurious due to the use of proxy cases.
- Authors used GCTCA-COJO for "fine-mapping", but GCTA-COJO is used to identify statistically independent signals and shouldn't be considered fine-mapping. The authors should perform Bayesian fine-mapping analysis (e.g., using susie) to better discern likely causal variants at specific loci.
- Authors should provide some metrics of the accuracy of imputation. For example, the authors can compare WGS-derived with imputed genotypes in samples with both data.
- The authors should provide more information about the STRs that are associated with AD. How many repeat units are there in the reference genome for each of these STRs? How many repeat units are the STR variant alleles? What are the allele frequencies of variant alleles for these associated STRs?
- The authors treat STRs as a biallelic genotype, which may introduce false associations. I recognize that most GWAS software are not equipped to handle multi-allelic genotypes. Nonetheless, for the identified STR disease associations, it would be helpful to repeat the analysis by treating the STR length as a quantitative variable.
- Overall, the manuscript is lacking in figures. In particular, it would be helpful to illustrate with locus zoom-type plots the associations at ABCA7 and APOE loci. Figures S5-18 showing these genomic loci are very grainy, hard to interpret, and would benefit from at least adding in gene tracks. It would also be helpful to illustrate some of the putative functional mechanisms, such as the association at SNX32.

Reviewer #3

(Remarks to the Author)

Comments to Gmelin et al.

The authors performed an analysis to investigate the missing heritability of Alzheimer's disease (AD) risk by conducting a genome-wide association study (GWAS) on short tandem repeats (STRs) using over 300,000 samples from the UK Biobank (UKB).

The methodology uses either directly called STRs or STRs inferred from SNP data using panels developed and published in other papers, primarily based on work by the Gymrek group, who are world-leaders in STR method development.

The authors identify over 100 STR genome wide significant signals, with many of these (reassuringly) tagging SNP haplotype, including many that tag the highly significant APOE locus.

Moreover, they identified 14 independent loci that have either been detected previously using SNP-based GWAS, make contributions to SNP-driven associations (i.e., APOE), or represent novel signals seemingly driven solely by STRs. The latter is one of the promises of this approach, since STRs have previously been shown to causally contribute to genetic risk of common traits.

The study identified two new potential STR-based signals for AD:

1. Near OVOL1/SNX32 on chromosome 11q13 (detected in imputed STR analysis)
2. Near WSB1 on chromosome 17q11 (detected only in the WGS-derived STR analysis)

These signals were not previously reported in SNP-based GWAS studies for AD, although the authors were not able to fully replicate these findings in their replication cohort.

For known AD loci, including ABCA7 and MINDY2/ADAM10, the authors identified nearby STRs that either represent the lead signal or make substantial contributions to SNP effects. Similarly, the authors report on STR associations in the HLA/MHC region and demonstrate that this signal is associated with DNA methylation and gene expression in human brain tissues.

Finally, the authors provide an estimate of the heritability of AD explained by STRs, calculating that they account for approximately 3% of the phenotypic variance in AD in this dataset.

This work represents one of the first comprehensive investigations into the effect of STRs on AD risk. The authors reference another paper (Guo et al., 2025; ref 19 in the paper) that also examined STRs in AD, albeit with a much smaller cohort (~3,000 samples) where only a signal in the well-established APOE region was identified. They directly call the genotypes of the STRs using similar methods underpinning the imputed dataset that Gmelin et al use.

The methods provide sufficient detail for the work to be reproduced, but we recommend including a link to a GitHub or Zenodo repository containing the analysis scripts to enhance reproducibility.

The authors should also consider making their STR GWAS summary statistics available through appropriate databases, which would provide substantial research value to the community. Current GWAS summary statistic catalogues are primarily geared toward SNPs, and this study presents an opportunity to encourage repositories such as EMBL-EBI to house STR GWAS results as well.

This manuscript presents innovative and comprehensive analyses of the role of STRs in AD genetics, identifying novel loci and demonstrating significant contributions of STRs to known AD risk regions. With appropriate revisions addressing the methodological concerns and reporting gaps highlighted below, this paper will make a valuable contribution to our understanding of the genetic architecture of Alzheimer's disease.

Specific suggestions for improvement

1. Effect size reporting for conditional analyses: The authors state that STRs make "substantial contributions" to known signals in regions like MINDY2/ADAM10 and ABCA7, or that certain STRs "represent the lead signal." Effect sizes (beta coefficients) for these conditional analyses are not adequately reported in the main text, making it difficult to ascertain the magnitude of the associations and their biological significance. While some of this information appears in supplementary tables, comprehensive reporting of effect sizes, standard errors, and confidence intervals would strengthen the manuscript.
2. Proxy phenotyping methodology: The authors derive a risk score for AD by categorizing cases based on either ICD codes or self-reported parental AD status. Several questions arise:
 - a. What is the UKB data field that describes the parental AD status question?
 - b. Have the authors validated this approach using related individuals in the UKB (where trios or parent-child pairs exist)?
 - c. How are the results affected if analysis is restricted to ICD-diagnosed cases only?
 - d. Given the method differs from the Jansen et al. paper, what evidence supports that the derived score accurately captures AD risk?
3. Age as a confounding factor: The GWAS model does not account for age in the linear models, which is surprising given that age is the strongest risk factor for AD (and is used in the derivation of the AD by proxy score). This omission could significantly impact the findings.
4. Replication approach: The replication cohort appears to consist of excluded related individuals from the "Other-White" subset, which is not truly an independent cohort. This approach should be explicitly explained and its limitations discussed.
5. Multi-allelic STR treatment: The decision to code multi-allelic STRs as biallelic is mentioned only in the discussion, despite being a critical analytical choice that affects interpretation. This methodological decision should be detailed earlier in the paper. We know that STR size influences many neurological disorders, so binarizing STR calls results in information loss. This approach makes it difficult to assess effects when treating alleles that are smaller or larger than the most common as equal, as they would be expected to have different or even opposing effects.
6. STR imputation vs. WGS-derived genotypes: Additional details on the concordance between imputed and WGS-derived STRs would strengthen the paper. While Supplementary Figure 21 shows good correlation between effect sizes, information on the concordance of the actual STR calls would be valuable. Moreover, the authors use predefined STR data from Halldorsson et al. (2022), but there is raw WGS data available for all UKB participants, from which STRs could be called using methods like ExpansionHunter, ExpansionHunterDenovo, or STRling. This could serve as an additional QC step for validating the imputed STRs and potentially increase the power of the analysis, particularly given that only one WGS-derived STR (near WSB1) reached genome-wide significance outside the APOE region.
7. Sample size and power: The requirement to use only unrelated samples results in a loss of power due to reduced sample size. Current generalized mixed models used for incorporating related individuals in SNP data analyses (e.g., SAIGE and REGENIE) are not currently applicable to multi-allelic marker data, but this limitation should be discussed.
8. Comparisons with previous work: The paper would benefit from a more detailed comparison with the Guo et al. (2025) study, including an analysis of concordance between results.
9. APOE region analysis: The authors highlight MHC signals in the discussion but provide less focus on the APOE signal despite analyzing it in depth. A graphic annotating the APOE region with all relevant markers, illustrating relative physical map positioning, LD (r^2), and types of STRs would be helpful, as one of the key messages is the increased complexity in the APOE region.
10. Sex-stratified analysis: A separate male and female analysis should be conducted, in line with Nature Communications reporting guidelines.

Line by line revisions

1. Line 127: The replication study should employ Bonferroni correction (0.05/14, not 0.05).
2. Terminology: Replace "descent groups" with "ancestry groups" throughout the manuscript.
3. P7 Line 143: Include the p-value for the COJO analysis of ABCA7 results.
4. P7 Lines 149-155: Clarify whether these loci have been previously examined for multiple SNP signals, and whether conditioning took into account additional signals.
5. P8 Line 168: Replace "dropped above suggestive significance" with "dropped, but were still showing suggestive association signals."
6. P10 Line 250: Clarify what is meant by "matching variants" and provide more details on the WGS-derived and imputed STRs with $p < 1e-05$.
7. P10 Line 250: Consider conducting concordance analyses with the results from Guo et al.

8. P10 Line 258: Define what is meant by "highlighted" STRs.
9. P10 Lines 219-231: Include contingency tables illustrating these results in supplementary materials.
10. P16 Line 371: Correct "decent" to "descent".
11. Figures 1 and 2: Describe in the legend how the AD and AD-by-proxy results were combined to generate single GWAS results.

Reviewer #4

(Remarks to the Author)

Version 1:

Reviewer comments:

Reviewer #1

(Remarks to the Author)

I appreciate the modifications. The revised manuscript shows substantial improvement over the initial submission. I have a few additional comments on reviewer responses and the updated results.

1. Please review the interpretation of the eQTM. The current results shown in Figure 3 indicate that the STR is linked to decreased DNAm, which in turn leads to lower SNX32 expression. However, DNAm sites are expected to silence gene expression, so the expected direction should be the opposite. The authors must try to resolve this ambiguity and adjust their mechanistic interpretation accordingly. If there is no additional evidence to support this effect, please acknowledge caution when interpreting such an unexpected direction of effect.

2. The responses for the number of PCs used to adjust the eQTL and mQTL analysis may not adequately control for confounders. The authors should either: (a) demonstrate that their PC choices are optimal by testing a range of values and showing that their chosen numbers yield the best trade-off between power and control (e.g., as is done in GTEx analyses), or (b) explicitly acknowledge in the Methods and Results that the PC choices were not optimized and that this represents a limitation, particularly for the SNX32 interpretation.

3. In Response 1.17, the authors address my concern about the robustness of heritability estimates derived from random subsets by rerunning the analysis across 17 independent batches. They report a mean STR heritability estimate of 2.6% but acknowledge that "heritability estimates from each run varied." I ask the authors to report the range and/or standard deviation of the 17 estimates in the Results section, in addition to the mean. If the variance across runs is large relative to the mean estimate, this should be discussed as a limitation.

Reviewer #2

(Remarks to the Author)

The authors have adequately addressed all of my concerns.

Reviewer #3

(Remarks to the Author)

The authors have substantially improved reporting of effect sizes and standard errors for conditional analyses at key loci (ABCA7, MINDY2/ADAM10), now presented both in-text and via new main-text figures (Figures 2C, 2D). The expanded concordance analysis between imputed and WGS-derived STRs which now covers ~622,000 STRs rather than only the 84 manually curated variants provides additional evidence that the imputation approach is sound. Moreover, the sensitivity analyses for age adjustment and proxy-case adequately demonstrate robustness to these analytical choices. The Bonferroni correction for replication ($p < 0.05/14$) has been appropriately applied.

The targeted multi-allelic STR analyses, performed in response to concerns about information loss from biallelic coding, confirmed 12 of 14 genome-wide significant loci, strengthening confidence in the primary findings. The sex-stratified GWAS and genotype-by-sex interaction analyses found no strong evidence for sex-specific effects satisfying our request. The concordance with Guo et al. at the APOE locus ($r^2 = 0.75$, $D' \approx 1$ between their respective lead STRs, separated by only 14bp) is reassuring. We accept the authors' justification for not producing a detailed APOE fine-mapping figure as this is probably out of scope for this manuscript.

The analysis code is now available on GitHub and summary statistics have been deposited on Zenodo as requested in our original review.

The remaining limitations, specifically the reliance on imputed STRs, restriction to European-ancestry samples and small functional validation sample, were acknowledged by the authors in the review comments and don't detract from the overall results.

The claim of inability to perform de novo STR calling from WGS to compare the imputed STRs to due to computational cost (Rebuttal point 3.6) requires a counterpoint. The authors claim about cost suggest that they thought that an evaluation across all STRs and individuals is required for this, but that is not the case. This could be performed with a random sample of 100 individuals across 100 randomly chosen loci. If the authors are not familiar with the tools or working in RAP this would have been a hurdle from an implementation point of view, but it is not a cost hurdle, as the authors claim, especially since UKBB RAP give compute credit in the first instance anyway. However, whilst desirable there is enough other supporting evidence, especially in the revision, and previous work, to render this not a strict requirement and the paper is acceptable in its current state.

Reviewer #4

(Remarks to the Author)

We thank all reviewers for their overall very positive appraisal of our work and their constructive comments and feedback. By running the requested additional analyses and by addressing the reviewers' concerns and suggestions, we feel that our manuscript was substantially strengthened and improved. All changes made with respect to the original submission are "tracked" in the revised version. In addition, we corrected occasional typos and grammatical errors, which are also tracked. In addition to the tracked version, we also provide a "clean" (untracked) copy of the revised manuscript. Below, please find a detailed point-by-point response to all comments raised by the reviewers.

REVIEWER COMMENTS

Reviewer #1 (Remarks to the Author):

In the paper entitled "GWAS on short tandem repeats identifies novel genetic mechanisms in Alzheimer's disease", the authors describe results of genome-wide associations of bi-allelic STRs with AD, for sample sizes reaching up to 330,000 individuals from the UK biobank. Tests were performed using imputation-derived STRs and, separately, WGS-derived STRs. As a result, authors identified 14 genome-wide significant hits using imputed calls (higher sample sizes) and 1 hit using WGS-derived calls. The authors also tested for replications in an independent set of individuals, showed the additional effects of STR to AD heritability, and explored potential functional implications of their findings via eQTL and meQTL mapping.

Comment 1.0: *Results regarding the implication of STRs in AD are relevant and much appreciated by the field. The methodology applied in this study was adequate. However, the overall presentation is subpar to what readers might expect in Nat. Communications. Many possibly interesting results are buried in supplementary materials, and the description of results can be improved to better highlight the strengths and findings.*

Response 1.0: We thank the reviewer for his/her thorough review of our manuscript and the overall positive evaluation and feedback. Furthermore, we appreciate your suggestions regarding clarity and presentation of the results. In response, we have substantially revised many parts of the manuscript to better highlight key findings and moved several important results from the supplement to the main text. Highlights of these edits include:

1. Now, the results section starts with a summary describing how the paper will flow.
2. The summary of the conditional analyses was moved from the Supplementary Material to the main text (Figures 2A-D).

3. Revised figures on our main findings with ABCA7 were also moved from the Supplementary Material and/or added newly to the main text (Figure 2E and 2F).
4. A new figure on our main findings with SNX32 was added (Figure 3).
5. A new table summarizing our main GWAS results was added (Table 1).
6. All results on WGS-derived STRs are now presented after the results on imputed STRs in the respective paragraphs.

We hope that these substantial changes go in the direction the reviewer had envisioned.

Comment 1.1: *The assumptions of findings resulting from imputed- and WGS-derived STRs are quite distinct. GWAS of imputed-STRs will inform if known GWAS SNP hits are in LD with STRs, which will be informative for fine-mapping. Although the conditional associations used in the manuscript are relevant, I believe that applying functional fine-mapping strategies has a better chance of capturing the real causal variants.*

Response 1.1: We thank the reviewer for raising this point, which was also highlighted by the other reviewer (see Comment 2.3 below). In response, we performed systematic fine-mapping of our top results using the summary statistics regression approach implemented in SuSiE (SuSiE-RSS). Furthermore, we deepened our analyses on the functional characterization of SNX32 (as requested by reviewer 2, see Comment 2.7) using brain-based DNAm and RNAseq data. These results of these novel analyses can be summarized as follows:

First, using the posterior inclusion probability (PIP) in SuSiE-RSS to select one lead variant per locus, the highest PIPs were mostly attained by SNPs, except for ABCA7, where an STR was identified as lead variant. While, formally, for SNX32 the highest PIP was also estimated for a SNP, PIP estimates for the lead STRs were also very high and only slightly smaller than for SNPs, indicating that STRs may contribute substantially to the increased AD risk in the region surrounding SNX32.

Second, and motivated by Comment 2.7 from reviewer 2, we also probed deeper into the relationship between STR alleles, DNA methylation and mRNA expression for our top hits. For our novel signal near SNX32, these analyses revealed that the STR allele increasing risk for AD in the primary GWAS is also associated with decreased DNAm at a nearby (intronic) CpG, which – based on newly performed eQTM (expression quantitative trait methylation) analyses in ~150 human entorhinal cortex samples – correlated with decreased SNX32 mRNA expression in brain. This is in agreement with prior work (e.g. Sugatha et al. 2023; ref. 37 in the revised submission) suggesting that downregulation of SNX32 leads to defects in neuronal differentiation in neuroglial cell lines. Our STR GWAS now provides an

interesting new molecular mechanism for the notion that less SNX32 mRNA increases the amount of AD pathology.

These and other findings from the adapted functional fine-mapping strategy are summarized in the Results (section “Discerning the drivers of STR-based GWAS signals”, first paragraph) while the eQTM methods were added to the Methods (section “Expression quantitative trait methylation (eQTM) analysis”) in the revised submission.

Comment 1.2: *Regarding the WGS-derived STRs, we can expect to find novel signals (with appropriate caution), considering that rare STRs are better captured and variants in more complex genomic regions are usually not well mapped in SNP arrays. Although the authors somewhat discuss these differences, I feel this should be more evident in the way the results are organized and presented. Currently, it looks as if the same analysis was just performed twice with two different sets of variants.*

Response 1.2: We agree with the reviewer in both aspects of this comment, i.e. that WGS-derived STRs can, indeed, be expected to better capture variants, especially those not well-mapped by SNPs and, second, that the analyses for imputed STRs and WGS STRs were essentially identical. We also agree with the reviewer that the description of our WGS-derived analyses (and their results) deserves more weight. Accordingly, we now lay one major emphasis on this point, see also details in Response 1.0 and the revised Results (section “Genome-wide association analyses on WGS-derived STRs”) in the revised manuscript).

Comment 1.3: *One important technical choice taken in the current paper was to split the repeats into bi-allelic variants, rather than to consider the actual lengths. Currently, it feels like this decision was made purely by convenience. There are obvious differences in considering either option, and both strategies could be complementary. For instance, by considering repeat lengths, you might capture effects of repeat expansions, while just bi-allelic variants probably represent repeats “fixed” in common haplotypes.*

Response 1.3: Again, the reviewer brings up an important point (also raised by reviewer 2, see Comment 2.6 below) which we have now carefully addressed in the revised submission. There were several reasons why we had initially decided to split multi-allelic STRs into bi-allelic variants, the two most important being: i) to benefit from the greater power of the full UKB sample by using imputed STRs, and ii) to increase comparability of our work with previous studies using imputed STRs (e.g. Bustos et al. and Ohlei et al.). However, the imputation catalogue contains a substantial number of multi-allelic STRs

that we detected only in a bi-allelic form. Notwithstanding, using only bi-allelic coding bears the risk of creating false-positive results at certain STR loci (a concern specifically emphasized by reviewer 2).

To particularly address this latter concern, we have now computed multi-allelic test statistics for all STRs showing genome-wide significant associations in the bi-allelic tests (summarized in new Supplementary Tables 5 and 6 and new Supplementary Figures 4 and 5). In summary, these multi-allelic analyses confirmed all previously identified STR loci except two, both located within the same 4.5Mb interval on chromosome 15q21-22, specifically near chr15:58846596 (MINDY2/ADAM10) and chr15:63139195 (LACTB). Conversely, statistical support also became stronger for some loci (e.g. chr1:207598421 [CR1] and chr17:63485772 [ACE]) in the multi-allelic analyses. The non-confirmation of the bi-allelic findings on chr.15 could be due to the possibility that the bi-allelic signal represents the effect of one allele that is “fixed” in a common haplotype while there is no effect of the other alleles at these STR loci, i.e. the situation suggested by the reviewer. We note, however, that this does not automatically invalidate the initial bi-allelic results in these regions as we identified other highly significant multi-allelic signals with other STRs that mapped into the same loci (Supplementary Table 6).

We have added all details regarding these novel analyses to the Methods (section “Multi-allelic Analysis”), Results (section “Genome-wide association analyses on imputed STRs”, last paragraph; also into new Supplementary Tables 5 and 6), and the Discussion (seventh paragraph) of the revised manuscript.

Comment 1.4: *The present study is very closely related to two papers referenced by the authors (both published in Nat. Communications). PMC11614882: “A phenome-wide association study of tandem repeat variation in 168,554 individuals from the UK Biobank”; and PMC11775329: “Polygenic burden of short tandem repeat expansions promotes risk for Alzheimer’s disease”. Both studies considered repeat lengths rather than genotyping alleles, besides other differences. The authors should make an effort to at least compare findings with previous related studies. This not only helps to understand the strengths and weaknesses of each approach but also strengthens the evidence for their main findings.*

Response 1.4: To address this comment, we have extended the discussion in our revised manuscript (sixth paragraph) and now provide a more detailed comparison of our findings vs. the two papers highlighted by the reviewer. In summary, we now clarify that while Manigbas et al. (PMC11614882) used a comparatively large, and partially overlapping subset of the UKB cohort (~170K individuals), their study was limited to a relatively small set (~36K) of high-quality polymorphic STRs. This may explain why the authors did not

identify any genome-wide significant signals in their analyses on AD risk. In contrast, the study by Guo et al. (PMC11775329) performed association testing on a comparatively large number of STRs (~300K) but only used a relatively small number of samples (~3K). Possibly because of the small sample size, Guo et al. only detected genome-wide significant associations with STRs in the APOE region (which were comparable to our findings near APOE, please see also our response to comment 3.8 by reviewer 3). By implementing STR imputation, our study maximized power by using both a larger number of samples (~330K) and STRs (~3M variants, of which 335K were independent) than the previous papers. We hope that the now extended comparison across studies goes in the direction that the reviewer intended in his/her comment.

Comment 1.5: *The results section starts by describing the results of the first GWAS. Starting with a summary describing how the paper will flow might help readability. If possible, please provide some general information about the STRs identified before jumping into association tests. Again, make a clear separation of the purposes of each section. Please bring more details of the main findings to the main Figures. The current paper has only 2 main figures (2 Manhattan plots) that are ok to show, but they are basically illustrative.*

Response 1.5: We thank the reviewer for this important feedback on how to improve the legibility and presentation of our results. In part, this relates to his/her earlier point (Comment 1.0) to which we have already responded above. With respect to the details of this comment (i.e. presentation of main results), we confirm that we have now substantially revised the corresponding section for greater clarity. Specifically, relevant sections have been rearranged such that we first present all association results (both for imputed and WGS-derived STRs, applying both bi-allelic and multi-allelic analyses), followed by sections on replication, *in silico* fine-mapping, and the meQTL/eQTL analyses. We have also added new or revised figures on our main findings with ABCA7 and SNX32 (Figures 2 and 3), the summary of the conditional analyses (Figure 2) and inserted a new table summarizing our main GWAS results (Table 1).

Comment 1.6: *The current interpretation of the results is quite simplistic. There are many locus zoom plots in the supplementary material that could help, but they still need visualization improvements. Also, for comparison, plotting results of an independent AD GWAS of the same locus (aligned with STR results) will help to highlight cases where the signal was replicated.*

Response 1.6: This comment extends this reviewer's previous suggestions on how to improve clarity and structure of the results presentation and we are thankful for this specific and helpful feedback! In response we have, first, substantially updated all locus zoom plots now showing more details (i.e. gene-tracks, improved color-coding; Supplementary Figures 7-19) and have added a visual comparison (new Supplementary Figure 3) of our STR-GWAS results with those from the SNP-GWAS published by Bellenguez et al. These comparisons reveal that there were strong SNP-based signals in all of our top 14 STR regions (all at genome-wide significance except for the new STR locus on chr11:65810443 [SNX32]; see also our response to Comment 1.8). Lastly, we have added a new figure summarizing the conditioning results across all 14 genome-wide significant loci (Figure 2). We hope the reviewer agrees that the interpretation of the main results is now facilitated and improved with the help of these new display items.

Comment 1.7: *Genome-wide association analyses on imputed STRs*

Line 110: If this STR association is real, we should expect to observe SNP associations in the locus. I suggest checking the AD GWAS complete summary stats.

Response 1.7: The reviewer is correct in his/her assumption and this is precisely what we observe, i.e. in our SNP-based GWAS there is a strong SNP-based signal (rs12800497, chr11:65812291, $p=2.64e-8$; and Supplementary Figure 6) near the STR association at chr11:65810443.

Comment 1.8: *Discerning the drivers of STR-based GWAS signals*

Line 148: The ABCA7 result is possibly interesting; more details could be shown in the main figures.

Response 1.8: We fully agree with the reviewer that the STR-based result near ABCA7 is of particular interest and have added a dedicated figure on that finding to the main text of the revised version (Figure 2).

Comment 1.9: *Heritability estimates for STRs compared to SNPs*

Line 181: The authors found that heritability based on SNP resulted in ~36%, while using only STRs resulted in ~31%. This is quite a high estimate, which is probably explained by APOE alone. This should be acknowledged.

Response 1.9: We agree with the notion of the reviewer that a large portion of estimated AD heritability is explained by the APOE locus. In the literature, it is estimated that APOE $\epsilon 4$

explains ~25% of the total heritability (PMIDs: 26312828, 27595457). However, the main reason for the comparatively high heritability estimates in our data is likely our choice of method: specifically, we used the GCTA-LDMS algorithm which, in general, returns higher numerical estimates than the LDSC method that is also often used in SNP-GWAS studies (reviewed in PMID:37131074). Importantly, our GCTA-LDMS results are in very good agreement with estimates from other authors using the same method (PMID: 37131074). To avoid misinterpretation, now we substantially updated the Results section of the revised manuscript (section “Heritability estimates for STRs compared to SNPs”).

Comment 1.10: *Genome-wide association analyses on WGS-derived STRs*

Line 196: Unclear if “Halldorsson-White-British” is a subset of “White-British” or a totally different individuals.

Response 1.10: We apologize for not having been clearer on this aspect and have now elaborated on this point in the Results of the revised manuscript (section “Genome-wide association analyses on WGS-derived STRs”, first paragraph). In brief, the answer is: yes, the “Halldorsson-White-British” sample is a subset of the “White-British” sample.

Comment 1.11: *Line 199: Were less frequent STRs also considered? This is an advantage of using WGS-derived variants.*

Response 1.11: We thank the reviewer for this question! The short answer is: no, for this study we did not consider rare STRs. Our decision was based on the fact that low-frequency STRs were only available in a subset of the UKB cohort (i.e. the portion of the WGS data processed for STR genotyping in the Halldorsson study [“Halldorson-White-British” subsample]). However, with only 928 diagnosed and 13,759 proxy AD cases, the size of this subsample is very small. A power analysis for variants at MAF of 0.001 revealed that we have less than 30% power to detect genome-wide significant associations in the combined AD case dataset (NB: actual power will likely be smaller due to misclassifications of the proxy AD cases), which is why we selected against including low frequency STRs in the current study.

However, this does not mean that we consider rare STRs as uninformative or uninteresting, quite the opposite is the case: in another arm of the STaR-AD project, we are using *de novo* WGS to identify rare and potentially disease-causing STRs in a collection of multiplex AD families. In that study, we use UKB data to run customized STR genotyping for a select set of (very) rare candidate STR loci. Since the analyses of that arm of our project are still ongoing, we were not able to include them into the current manuscript (which is focused

on common STRs). Notwithstanding, we have now added this limitation to the discussion of the revised manuscript (discussion, seventh paragraph, end).

Comment 1.12: *Comparison of imputed vs. WGS-derived STRs*

Line 249: Why only 84 variants matched? It would be interesting to show the concordance of allele frequencies between strategies.

Response 1.12: Again, we apologize for not having been clearer on this issue and thank the reviewer for his/her thorough reading of our manuscript.

First, we would like to explain why matching STR variants reliably is challenging. Variants come from different toolkits and catalogs, leading to differences in their positions, and their allele sequences and how variants are named (i.e. there is no established ID for STRs like rs numbers for SNPs). As a result, there is no straightforward one-to-one mapping between variants, since imputation can produce different repeat units or slightly shifted positions, and even variants at the same position might not have matching alleles. Since manual look-up is not feasible for all variants, we only manually checked variants that passed at least suggestive significance ($p < 1e-5$) in both analyses. In total, for WGS-derived and imputed STRs this applied to $n=133$ and $n=482$ STRs, respectively. Intersecting these two sets for matches in all of the above criteria results in $n=84$ overlapping STRs.

We also thank the reviewer for suggesting to compare the allele frequencies between strategies, as this adds another layer of supporting evidence to our work. Moreover, motivated by comments of all reviewers, we now invested additional efforts and implemented a novel computational method that allows to perform matching of STR alleles by genomic position and by allele length at a much larger scale than in the original submission. Thus, we now report correlations for allele frequencies, effect size estimates (beta values), and sum of allele sizes between imputed and WGS-derived STRs both for all $n=84$ previously manually curated STRs, but also for $\sim 600K$ other STRs in the Results (section “Comparison of imputed vs. WGS-derived STRs” first paragraph, middle and second paragraph, beginning) and Methods (section “Comparison of GWAS results for imputed vs. WGS-derived STR variants”). We also added new Supplementary Figures 22-24 on this topic. Please also see our response to comments 2.4 and 3.6 from reviewer 2 and reviewer 3 below.

We note that STR variants in the $\sim 600K$ set are not curated for SNPs possibly mapping to repeat sequences, correct reference allele assignment, or small shifts in genomic position. In response to this comment, we also revised Supplementary Figure 21 to more directly display matching procedure.

Comment 1.13: *Delineating potential functional mechanisms of AD-associated STRs.*

Line 259: Not clear if DNAm and RNAseq were performed in the same individuals used in the GWAS. If it is an independent dataset, why was this particular one chosen? There are other sources of omics data with a better sample size that could be used for this purpose.

Response 1.13: We apologize for not having more clearly addressed the question of overlap between GWAS and DNAm/RNAseq data in the original submission. The short answer is: no, these datasets do not overlap. Moreover, the reviewer is correct that larger human brain datasets exist with DNAm or RNAseq data (the list is considerably shorter when only datasets are counted that have both DNAm *and* RNAseq data available). However, and most importantly, at least to our knowledge, these datasets do not have pre-computed STR genotype data, neither imputed nor WGS-derived, which is the molecular “anchor point” in linking the UKB-based STR GWAS results to DNAm or RNAseq in our study. This is different for the OPTIMA dataset utilized here, for which we had applied the same exact STR imputation pipeline also used for the UKB data and had ready access to precomputed STR-meQTL and STR-eQTL statistics performed in other projects of our group. We have added a short justification highlighting these points on the usage of the OPTIMA brain dataset to the Methods of the revised submission (section “Methylation quantitative trait locus (meQTL) analysis”).

In addition to these STR-based meQTL and eQTL analyses, we have now also performed eQTM analyses, where we directly correlate levels of DNAm and mRNA expression for all CpGs showing FDR-significant associations with top STRs. These analyses, confirm the positive correlation between DNAm and mRNA expression of various HLA isoforms and SNX32 supporting the overall conclusions of the STR-based meQTL and eQTL analyses. For more details on these latter findings, please see also our response to comment 2.7 from reviewer 2 below.

Comment 1.14: *Methods*

Line 518. What threshold of kinship was used?

Response 1.14: We used a threshold of 0.025 in PLINKs KING kinship coefficients (--king-cutoff 0.025) which corresponds to removing relatives up to the fourth-degree (second cousins). This important methodological detail has now been added to the Methods of the revised manuscript (section “Human subjects”, first paragraph, end), we apologize for the oversight of not providing it in the original submission.

Comment 1.15: *Line 538. This strategy was derived by Jansen et al. Do other AD GWAS apply similar strategies?*

Response 1.15: This comment relates to the “proxy phenotyping” to increase the number of AD cases available in the primary GWAS. The general method was actually developed earlier (by Liu et al, Nat Genet 2017) and then implemented widely for the first time in Jansen et al. To our knowledge, most AD GWAS published after the Jansen et al. paper using the UKB or other biobanks used similar implementations of proxy phenotyping for inclusion of comparatively young biobank participants. For a more detailed discussion on our proxy-phenotyping please also see our answers in response to comment 3.2 and 2.2 for sensitivity analysis without proxy-phenotypes.

Comment 1.16: *Line 560. By splitting multiallelic STRs into bi-allelic variants, the information regarding potential repeat expansions is lost. So, basically, the STRs surviving QC are “fixed” insertions of repeat patterns. Please include some considerations regarding these underlying conditions.*

Response 1.16: Again, we thank the reviewer for pointing this out. We believe that this comment falls under the general topic “limitations of the STR imputation approach”, already discussed elsewhere in this point-by-point response (e.g. our responses to comments 1.2 and 1.3, see above). Furthermore, we have now implemented multi-allelic analyses for all genome-wide significant bi-allelic signals and confirmed that the vast majority also show very strong associations using this approach (response 1.3).

Comment 1.17: *Line 636: Randomly subsetting of individuals raises questions about the robustness of the heritability estimates. Please show that this is not biased.*

Response 1.17: We agree with this comment that sub-setting may invalidate the robustness of the heritability estimates and have therefore substantially extended the analyses in this arm of our study. As mentioned in the original submission, the sub-setting was necessary because of computational constraints since calculation of the pairwise GRM relationship matrix for approximately 300,000 samples requires hundreds of GB of RAM. For these reasons, we are still required to use sub-setting, but for the revised submission have now analyzed the full UKB dataset in this way. Specifically, we reran the heritability estimates using *all* AD cases combined with random subsets of all controls (N=249,681), i.e. 17 batches of ~15,000 samples each. Although, as expected, heritability estimates from each run varied, the mean estimate (2.6%) for the full dataset was identical to the estimate derived from just 1 subset in the original submission. The results of all

heritability runs have now been included in a new Supplementary Table 19, and the corresponding sections of the revised manuscript have been edited accordingly in the Results and Methods (sections “Heritability estimates for STRs compared to SNPs” and “Heritability Analysis”, respectively).

Comment 1.18: Line 645: *Although it’s not a straightforward task to compare different STR calls, using different IDs as an explanation is not adequate.*

Response 1.18: We fully agree and apologize for not having been clearer on this issue. In many ways, this issue is addressed in comment 1.12, so please see our response to this critique above.

Comment 1.19: Line 648: *Given the simplistic condition used to match variants, please provide a comparison of allele frequencies of matched STRs before and after manual curation.*

Response 1.19: Following this suggestion we have now calculated and plotted correlations of allele frequencies between matched STRs before (left) and after (right) manual curation (see figure below). Motivated by this reviewer’s comments, we have also improved our STR matching procedure, i.e. alleles are now algorithmically (and no longer manually) matched by position and by their length. Notwithstanding these improvements, manual curation is still needed in case of genomic position shifts, presence of sequence variants, and switched assignment of reference and alternative alleles, which was done for the n=84 variants on the right.

In the Methods of the revised manuscript, we have now substantially extended the section on allele matching (section “Comparison of GWAS results for imputed vs. WGS-derived STR variants”) and Results (section “Comparison of imputed vs. WGS-derived STRs”, first paragraph) and added the frequency correlation plot for comparable variants to the supplement (as new Supplementary Figure 23).

Comment 1.20: *Line 653: What is meant by targeted meQTL analysis?*

Response 1.20: We apologize for not having been clearer on defining this term. By “targeted meQTL analysis” we mean the search for STR-based methylation quantitative trait loci, i.e. STRs associated with DNA methylation, for CpGs located in the immediate vicinity (i.e. ± 1 Mb) of the genome-wide significant STRs emerging from the main GWAS analyses. So: the term “targeted” refers to the specific set of CpGs up- and downstream of the top GWAS STRs. To make this clearer to potential readers of our manuscript, we have reworded the relevant Methods section for the revised submission (section “Methylation quantitative trait locus (meQTL) analysis”).

Comment 1.21: *Lines 660 and 662: The DNAm data analyzed was generated by Sommerer et al., while the QTL methodology followed Ohlei et al.? If that is the case, what is the ms in preparation (line 665) about? The text is not clear on what was done in the present study vs. what was obtained.*

Response 1.21: A first description of all methods related to the *generation and processing* of the DNAm data is summarized in Sommerer et al, while the general methodology to *perform meQTL analyses* in this dataset using SNPs (not STRs) was described in Ohlei et al. The manuscript in preparation (Happel et al.) reports the results of meQTL analyses using STRs (not SNPs) and represents, to our knowledge, one of very few studies to calculate this in human brain, and the first of its kind for entorhinal cortex. For the 14 STRs highlighted by our GWAS, we looked up the corresponding STR-based meQTL results from that latter study. We hope to be ready to submit the Happel et al. manuscript in early 2026. We have now reworded the corresponding Methods paragraph to avoid confusion (section “Methylation quantitative trait locus (meQTL) analysis”).

Comment 1.22: *Line 664: Were other numbers of DNAm PCs tested? Usually, this improves meQTL detection.*

Response 1.22: No, we only tested the first three DNAm PCs in the final meQTL analyses of this study. This number was determined based on the following scree plot.

We hope the reviewer agrees that selecting the first three DNAm PCs is an appropriate choice in this context.

Comment 1.23: Line 668: *What is meant by targeted eQTL analysis?*

Response 1.23: This terminology follows the same logic as “targeted meQTL analysis” explained above (Comment 1.20). To make this clearer to potential readers of our manuscript, we have reworded the relevant Methods paragraph for the revised submission (section “Expression quantitative trait locus (eQTL) analysis”).

Comment 1.24: Line 671: *I understand this is re-using data, but a brief description is welcome.*

Response 1.24: We agree and now provide more details for this arm of our study in the corresponding Methods paragraph in the revised manuscript (section “Methylation quantitative trait locus (meQTL) analysis”).

Comment 1.25: Line 674: *Were other numbers of RNA PCs tested? Usually, this improves eQTL detection.*

Response 1.25: Again, in these analyses we only tested the scenario including the first two PCs in the final eQTL computations. This number was determined based on the following scree plot:

Again, we hope the reviewer agrees that selecting the first two RNA PCs is an appropriate choice capturing most of the in this context.

Comment 1.26: *Line 676: Please make summary statistics publicly available.*

Response 1.26: We fully agree and have deposited all relevant summary statistics with public availability on zenodo.org (<https://doi.org/10.5281/zenodo.17908177>).

Comment 1.27: *Line 683: Please make the code used for analysis publicly available.*

Response 1.27: Again: we fully agree and thank the reviewer for these suggestions. Accordingly, we have now created a github page with all relevant code used for the analyses in the current manuscript (URL: https://github.com/dgmelin/ukb_ad_str_gwas). This URL has also been added to the revised manuscript (“Code availability”). Furthermore, a zipped archive of all scripts was added to zenodo.org (same link as the summary statistics, see above).

Reviewer #2 (Remarks to the Author):

Comment 2.0: *In this manuscript, Gmelin and colleagues perform genome-wide associations between STRs and Alzheimer's disease risk in the UK Biobank. Their work assessed a large number of STRs genome-wide using both imputed STR genotypes and WGS-derived genotypes. They identify a number of STR disease associations, many of which are at known AD GWAS loci. At several loci, the STR associations appear to be at least partially conditionally independent from known SNP associations, suggesting that the STRs may have a causal role at the loci. Their work extends upon recently published findings linking STRs with AD risk.*

Response 2.0: We thank this reviewer for his/her positive and insightful summary of our work. In particular, we are grateful you recognized the scale and novelty of our STR analysis and its contribution to understanding AD genetics and we greatly appreciate your recognition of our findings and their potential implications.

However, there are a number of major issues that need to be addressed:

Comment 2.1: *The use of a biobank results in an unbalanced case/control ratio which can lead to skewed test statistics. Saddlepoint approximation and other methods have been used in these scenarios to reduce Type I error. The authors should demonstrate that their associations are not the result of a highly unbalanced case/control ratio.*

Response 2.1: We thank the reviewer for raising this point, which relates to essentially every case-control GWAS run using UKB data. With respect to our analyses, we note that saddlepoint approximation was originally developed for studies with highly imbalanced case/control ratios, which - in our opinion - is not a predominant problem in our study: First, our case-control ratio is approx. 1:5 when contrasting the ~46K cases (43K proxies + 3K diagnosed) to ~250K controls and this is typically not considered "highly unbalanced". Second, we treat case status as a quantitative phenotype and, hence, do not compute standard bi-variate case-control statistics but instead use a linear model.

Notwithstanding, motivated by this comment, we were interested in how our results would change when applying the saddlepoint estimation suggested by the reviewer and, hence, ran two types of additional analyses using dichotomized phenotypes: First, we grouped all cases (diagnosed and proxy) into one group allowing us to run standard case-control analyses via a logistic regression model. To achieve this, we used the Regenie software, which has a function implemented to run this type of analysis. The Pearson correlation coefficient of effect sizes (beta values) for association statistics with and without

saddlepoint estimation indicates a nearly perfect correlation for all variants ($R=0.9998$). This correlation became even stronger when considering only those variants showing at least suggestive evidence of association in the original GWAS ($n=482$, $R=0.99992$). These results clearly show that comparing logistic regression results with or without including saddlepoint approximation does not appreciably influence the results, supporting our initial notion that our study is not generally affected by a “highly unbalanced” design.

Since the analyses of our primary GWAS used a linear model, we decided against including the details of these add-on analyses into the revised manuscript in the interest of space and to not confuse potential readers of our work. However, we would be happy to re-revise our manuscript in this regard (and include all the details of the saddlepoint analyses and results) should the reviewer deem it necessary.

Comment 2.2: *Recent work has suggested potential biases in GWAS by proxy for Alzheimer’s disease (see PMID: 39496879). The authors should demonstrate that their results are not spurious due to the use of proxy cases.*

Response 2.2: We thank the reviewer for this comment and note that we had commented on this topic briefly in the discussion of our original submission where we wrote:

“Third, the proxy phenotyping approach used was recently shown to be the source of some bias in the context of GWAS, mostly by potentially inflating or skewing estimates of genetic correlations between traits⁴³. It is important to emphasize that only a negligible bias was detected for the discovery of GWAS loci⁴³, which is how proxy phenotyping was used here.”

Notwithstanding and motivated by the comments from this reviewer and reviewer 3 (Comment 3.2), we have now rerun all GWAS analyses only on diagnosed AD cases ($n=2,947$) and all controls ($n=249,681$) and correlated effect sizes from these analyses with those from our original GWAS (where we had also included $\sim 43K$ proxy AD cases). While, as expected, we lose statistical power due to the much smaller case sample, effect sizes (i.e. beta-values) correspond well across these analyses: they show a Pearson correlation coefficient of 0.8 when including all variants; this increases to 0.99 when only including those $n=482$ variants showing at least suggestive significance in the original GWAS. Hence, the results of our paper – in particular those showing at least suggestive evidence of association – remain largely unaffected by the use of proxy cases. Accordingly, we have appended the following sentence to the above statement in the discussion of the revised manuscript:

“This is also what we observe in our study where effect sizes (beta values) of GWAS analyses computed without the $\sim 43K$ proxy AD cases correlated highly with the results of

the primary GWAS (Pearson's $r=0.8$ for all and $r=0.99$ for at least suggestively significant variants)."

Comment 2.3: *Authors used GCTA-COJO for "fine-mapping", but GCTA-COJO is used to identify statistically independent signals and shouldn't be considered fine-mapping. The authors should perform Bayesian fine-mapping analysis (e.g., using susie) to better discern likely causal variants at specific loci.*

Response 2.3: We agree with this comment, which overlaps with a similar suggestion made by reviewer 1. Thus, please see our response to Comment 1.1 above for details on how we have implemented Susie in the revised manuscript.

Comment 2.4: *Authors should provide some metrics of the accuracy of imputation. For example, the authors can compare WGS-derived with imputed genotypes in samples with both data.*

Response 2.4: We agree with the reviewer that this type of analysis would improve our manuscript and have now added different metrics on the accuracy of imputation. This applies to two approaches used for matching variants. First, there is the initial approach of manually matching variants which minimally passed suggestive significance in both the imputed and WGS-derived datasets ($n=84$). Second, motivated by comments of all reviewers, we now invested additional efforts to develop and apply a novel computational method to perform the matching of STR alleles by genomic position and by allele length at large-scale ($n\sim 600k$). Spearman correlations of sum of allele lengths per locus showed a mean correlation of $r=0.89$ and mean $r=0.91$ for the sets of $n\sim 600k$ and $n\sim 84$, respectively. For the set of $n\sim 600k$, r^2 -values are greater than 0.8 for 84% of STR-loci and greater than 0.9 for 48%. While these numbers are slightly lower than in equivalent metrics published in Margoliash et al. ($r^2>0.9$ for 78.7% and $r^2>0.8$ for 92.7%) we note that our analyses are based on a substantially larger pool of variants, i.e. $\sim 600K$ STRs vs. only 408 in Margoliash. For more details on this topic, please also see our responses to comments 1.12 and 3.6 from reviewer 1 and reviewer 3, respectively, and the revised Methods (section "Comparison of GWAS results for imputed vs. WGS-derived STR variants") and Results (section "Comparison of imputed vs. WGS-derived STRs").

Comment 2.5: *The authors should provide more information about the STRs that are associated with AD. How many repeat units are there in the reference genome for each of*

these STRs? How many repeat units are the STR variant alleles? What are the allele frequencies of variant alleles for these associated STRs?

Response 2.5: We fully agree with the reviewer that more information about associated STRs would be interesting, including information on how many repeat units are in the reference genome for each associated STR. Hence, in Supplementary Table 3, we have now replaced entries in the columns “Reference allele”, “Alternative allele”, and “Allele 1” with repeat unit values. In the same table, we have also added an additional column to this table reporting “Frequency (Allele 1) in UKB dataset”.

Comment 2.6: *The authors treat STRs as a bi-allelic genotype, which may introduce false associations. I recognize that most GWAS software are not equipped to handle multi-allelic genotypes. Nonetheless, for the identified STR disease associations, it would be helpful to repeat the analysis by treating the STR length as a quantitative variable.*

Response 2.6: We thank the reviewer for raising this issue in his/her review. In response, we have now run multi-allelic tests for STRs showing genome-wide significance in the bi-allelic analyses to assess the possibility of false-positive associations. These analyses revealed a very good correspondence in the bi- vs. multi-allelic test statistics. For more details, see our response to comment 1.3 from reviewer 1, who had asked a similar question.

Comment 2.7: *Overall, the manuscript is lacking in figures. In particular, it would be helpful to illustrate with locus zoom-type plots the associations at ABCA7 and APOE loci. Figures S5-18 showing these genomic loci are very grainy, hard to interpret, and would benefit from at least adding in gene tracks. It would also be helpful to illustrate some of the putative functional mechanisms, such as the association at SNX32.*

Response 2.7: We thank the reviewer for pointing this out and suggesting ways to improve the overall aspect of our manuscript. Similar suggestions were also made by reviewer 1. In response to both reviewers’ comments, we have substantially revised and restructured our manuscript in the hope that it is now easier to grasp its full scope and scientific results of our study. Specifically, please see our responses to comments 1.0, 1.5, and 1.6 from reviewer 1 above.

With respect to the reviewer’s second suggestion, i.e. to better illustrate some of the putative functional mechanisms, such as the association at SNX32, we probed for potential correlations of DNAm levels at cg15531562 and the amount of SNX32 transcripts in brain samples derived from entorhinal cortex (OPTIMA) in newly performed eQTM

analyses. This revealed that increased levels of DNAm were correlated with increased levels of mRNA transcription. While increased DNAm is typically regarded as a repressor of gene transcription, the inverse effect direction observed here is consistent with data from the literature suggesting that increased DNAm within gene bodies can elicit non-canonical effects, i.e. increase gene transcription (e.g. PMIDs: 20133333, 34520760).

Based on these novel results, we believe that carrying the five-repeat allele at STR chr11:65810443:T(CA)_n is associated with decreased DNAm levels at cg15531562 (located in intron 1 of SNX32; beta = -0.0291, Supplementary Table 16). The reduced levels of DNAm will likely lead to a reduced expression of SNX32 mRNA, at least for transcript ENST00000308342 for which we directly observed such a correlation (Figure 3). These observations fit well to two recent functional studies (Wingo et al. 2021, 2025) showing that lower levels of SNX32 protein are causally associated with AD. Lastly, a third study (Sugatha et al. 2023) showed that lower levels of SNX32 protein negatively affect neurite outgrowth. A summary of these results and the suggested mechanism is now provided in Figure 3, which was newly added to the manuscript.

Reviewer #3 (Remarks to the Author):

Comment 3.0: *The authors performed an analysis to investigate the missing heritability of Alzheimer's disease (AD) risk by conducting a genome-wide association study (GWAS) on short tandem repeats (STRs) using over 300,000 samples from the UK Biobank (UKB).*

The methodology uses either directly called STRs or STRs inferred from SNP data using panels developed and published in other papers, primarily based on work by the Gymrek group, who are world-leaders in STR method development.

The authors identify over 100 STR genome wide significant signals, with many of these (reassuringly) tagging SNP haplotype, including many that tag the highly significant APOE locus.

Moreover, they identified 14 independent loci that have either been detected previously using SNP-based GWAS, make contributions to SNP-driven associations (i.e., APOE), or represent novel signals seemingly driven solely by STRs. The latter is one of the promises of this approach, since STRs have previously been shown to causally contribute to genetic risk of common traits.

The study identified two new potential STR-based signals for AD:

- 1. Near OVOL1/SNX32 on chromosome 11q13 (detected in imputed STR analysis)*
- 2. Near WSB1 on chromosome 17q11 (detected only in the WGS-derived STR analysis)*

These signals were not previously reported in SNP-based GWAS studies for AD, although the authors were not able to fully replicate these findings in their replication cohort.

For known AD loci, including ABCA7 and MINDY2/ADAM10, the authors identified nearby STRs that either represent the lead signal or make substantial contributions to SNP effects. Similarly, the authors report on STR associations in the HLA/MHC region and demonstrate that this signal is associated with DNA methylation and gene expression in human brain tissues.

Finally, the authors provide an estimate of the heritability of AD explained by STRs, calculating that they account for approximately 3% of the phenotypic variance in AD in this dataset.

This work represents one of the first comprehensive investigations into the effect of STRs on AD risk. The authors reference another paper (Guo et al., 2025; ref 19 in the paper) that also examined STRs in AD, albeit with a much smaller cohort (~3,000 samples) where only a signal in the well-established APOE region was identified. They directly call the genotypes of the STRs using similar methods underpinning the imputed dataset that Gmelin et al use. The methods provide sufficient detail for the work to be reproduced, but we recommend including a link to a GitHub or Zenodo repository containing the analysis scripts to enhance reproducibility.

The authors should also consider making their STR GWAS summary statistics available through appropriate databases, which would provide substantial research value to the

community. Current GWAS summary statistic catalogues are primarily geared toward SNPs, and this study presents an opportunity to encourage repositories such as EMBL-EBI to house STR GWAS results as well.

This manuscript presents innovative and comprehensive analyses of the role of STRs in AD genetics, identifying novel loci and demonstrating significant contributions of STRs to known AD risk regions. With appropriate revisions addressing the methodological concerns and reporting gaps highlighted below, this paper will make a valuable contribution to our understanding of the genetic architecture of Alzheimer's disease.

Response 3.0: We are grateful to the reviewer for his/her comprehensive evaluation of our manuscript and for their positive and very constructive feedback. We particularly appreciate their recognition of the study's key findings, originality, and relevance to the AD field!

Regarding the deposition of code and summary statistics, we fully agree and have now deposited all relevant summary statistics on the Zenodo database (<https://doi.org/10.5281/zenodo.17908177>). We have also created a github repository with all relevant code used for the analyses in the current manuscript (URL: https://github.com/dgmelin/ukb_ad_str_gwas). This URL has also been added to the revised manuscript (section "Code availability"). Furthermore, all scripts have been compiled into a zipped archive which was also uploaded to zenodo.org (same link as the summary statistics, see above).

Comment 3.1: *Effect size reporting for conditional analyses: The authors state that STRs make "substantial contributions" to known signals in regions like MINDY2/ADAM10 and ABCA7, or that certain STRs "represent the lead signal." Effect sizes (beta coefficients) for these conditional analyses are not adequately reported in the main text, making it difficult to ascertain the magnitude of the associations and their biological significance. While some of this information appears in supplementary tables, comprehensive reporting of effect sizes, standard errors, and confidence intervals would strengthen the manuscript.*

Response 3.1: We thank the reviewer for this suggestion and have updated the manuscript to now display the effect sizes and standard errors for STR signals mentioned (section "Discerning the drivers of STR-based GWAS signals"). Furthermore, we added these numbers to the corresponding table in the supplement (Suppl. Table 9). Lastly, to facilitate the interpretation of the results from the conditional analyses, we have generated two additional comparison plots and added them to the main text (now Figures 2C and 2D). These new plots are similar in design to the original plots comparing statistical support (p-values) before and after conditioning (now Figures 2A and 2B in the revised manuscript).

The new plots visualize the change in effect sizes for SNPs and STRs before and after reciprocal conditioning. Although effect sizes drop for all variants after conditioning, the reduction is smaller for the STR variants in genes like ABCA7, MINDY2/ADAM10 or SNX32, indicating that sizeable residual effects persist. The only exception was observed for APOE where the strong effect of rs429358 (“epsilon-4”) remains even after conditioning on the top STR, whereas the effect size of the STR variant drops considerably when conditioning on rs429358.

Comment 3.2: *Proxy phenotyping methodology: The authors derive a risk score for AD by categorizing cases based on either ICD codes or self-reported parental AD status. Several questions arise:*

- a. *What is the UKB data field that describes the parental AD status question?*
- b. *Have the authors validated this approach using related individuals in the UKB (where trios or parent-child pairs exist)?*
- c. *How are the results affected if analysis is restricted to ICD-diagnosed cases only?*
- d. *Given the method differs from the Jansen et al. paper, what evidence supports that the derived score accurately captures AD risk?*

Response 3.2: We thank the reviewer for raising this issue in his/her review and would like to address these questions as follows:

a) Parental AD status is described in the UKB-data fields 20107 and 20110 as “Illnesses of father/mother”, respectively. This field contains a list of reported illnesses of which AD/dementia is coded as 10 (data-coding 1010). To make this clearer to potential readers of our paper, we have now also added this information to the methods section of the revised manuscript (section “Human subjects”).

b) The “proxy phenotyping” approach applied here was originally developed and validated by Liu et al. 2017, and subsequently applied to all major AD GWAS using UKB data (e.g. Jansen et al., 2019; Wightman et al., 2021; and Bellenguez et al., 2022). Here, we simply adopted this approach (in slightly revised form, see d) to our study. We have now also added this information to the methods section of the revised manuscript (section “Procedure for proxy phenotyping”).

c) In response to Comment 2.2. from Reviewer 2, we repeated the GWAS without proxy-cases only including AD patients diagnosed by ICD10. As expected, the resulting associations lose statistical support (due to the loss in power) but correlation analyses of the effect sizes (beta values) show a very substantial overall agreement (Pearson’s $r=0.8$ for all and $r=0.99$ for at least suggestively significant variants in the original analyses). We

have added one sentence to reflect the results of these sensitivity analyses to the discussion section of the manuscript.

d) The problem – at least in our opinion – with the approach chosen by Jansen et al. (and subsequent GWAS) is that they assign equal scores to ICD10-diagnosed AD cases and “top-probability” proxy cases (i.e. probands for whom both parents are reported to have suffered from AD). Our implementation assigns higher scores (=4) to ICD10 cases compared to top-probability proxy cases (= maximally 2) and, as a result, better reflects the clinical reality. We further clarified this in the revised manuscript (section “Procedure for proxy phenotyping”).

Comment 3.3: *Age as a confounding factor: The GWAS model does not account for age in the linear models, which is surprising given that age is the strongest risk factor for AD (and is used in the derivation of the AD by proxy score). This omission could significantly impact the findings.*

Response 3.3: While we agree that age is the most important risk factor for AD, we initially did not include it in our models as we do not consider this variable a potential confounder (which would represent the strongest justification to account for it as a covariate in the regression models). In epidemiological terms, a confounder must be associated with both the outcome (which is the case, as correctly pointed out by the reviewer) and the exposure. Since germline genotypes are fixed at conception, they are generally independent of age, meaning age may function here as a “precision” variable rather than a confounder.

Notwithstanding, motivated by the reviewer’s comment, we have now recalculated all GWAS analyses including age as a covariate as sensitivity analysis. As suspected, the inclusion of age had virtually no impact on the results evidenced by Pearson correlations of beta values before and after age adjustment: correlation coefficients are $r=0.9984$ when comparing all variants and $r=1$ when only considering at least suggestively significant variants.

In addition, we quantified the “gain in precision” resulting from including age as additional covariate by comparing the standard errors of the effect estimates for the lead variants at minimally suggestively significant loci ($n=41$). This comparison revealed almost no reduction in SE ratio (median reduction= $0.16\%-0.17\%$), again indicating that age does not have a noteworthy impact on the GWAS results.

In summary, including age as a covariate showed negligible impact on effect sizes, p-values, and precision, confirming our original decision to not include it as covariate in the context of this GWAS. Notwithstanding, we have now elaborated on our rationale

(Methods; section “Genome-wide association analyses”) and added the results of the correlation analyses (before and after adjustment for age) as sensitivity analysis to the revised results (section “Genome-wide association analyses on imputed STRs”) and supplementary material (Supplementary Figure 25, Supplementary Table 20).

Comment 3.4: Replication approach: The replication cohort appears to consist of excluded related individuals from the "Other-White" subset, which is not truly an independent cohort. This approach should be explicitly explained and its limitations discussed.

Response 3.4: We apologize for not having been clearer on this aspect and have now elaborated on this point in the Methods. Actually, and importantly, the “Other-White” subset is entirely independent from the primary GWAS sample. The terms “White-British” and "Other-White" are assigned to subsets of the self-defined “White” cohort by UKB Data Field 21000. Based on this information, we split the cohort into “White-British” samples and, for replication, combined all remaining “White” samples (i.e. “Any other white background” and “White-Irish”) into our subset of “Other-White” individuals. To make this clearer to potential readers of our paper, we have now extended the description of this procedure in the revised version of our manuscript (Methods, section “Human Subjects”).

Comment 3.5: *Multi-allelic STR treatment: The decision to code multi-allelic STRs as biallelic is mentioned only in the discussion, despite being a critical analytical choice that affects interpretation. This methodological decision should be detailed earlier in the paper. We know that STR size influences many neurological disorders, so binarizing STR calls results in information loss. This approach makes it difficult to assess effects when treating alleles that are smaller or larger than the most common as equal, as they would be expected to have different or even opposing effects.*

Response 3.5: We thank the reviewer for raising this important point. First, we would like to clarify that the biallelic splitting did not binarize STR calls but treated each allele size at the given loci as a combination of observed reference and alternative alleles (as described in Bustos et al (2023)). However, the other two reviewers also raised concerns that bi-allelic analyses may have biased our top results, which prompted us to run multi-allelic tests for STRs showing genome-wide significance in the bi-allelic analyses. In brief, the multi-allelic analyses confirmed all originally identified STR loci except two, both located within the same 4.5Mb interval on chromosome 15q21-22. All other loci were supported in the multi-allelic analyses sometimes even showing stronger statistical support than in the

bi-allelic results. For a detailed account of the results from these analyses and where they were added to the revised manuscript, please see our detailed response to Comment 1.3.

Comment 3.6: *STR imputation vs. WGS-derived genotypes: Additional details on the concordance between imputed and WGS-derived STRs would strengthen the paper. While Supplementary Figure 21 shows good correlation between effect sizes, information on the concordance of the actual STR calls would be valuable. Moreover, the authors use predefined STR data from Halldorsson et al. (2022), but there is raw WGS data available for all UKB participants, from which STRs could be called using methods like ExpansionHunter, ExpansionHunterDenovo, or STRling. This could serve as an additional QC step for validating the imputed STRs and potentially increase the power of the analysis, particularly given that only one WGS-derived STR (near WSB1) reached genome-wide significance outside the APOE region.*

Response 3.6: We agree with the overall notion of this critique, which overlaps with comments from the other reviewers (i.e. see Comments 1.12 and 2.4. above). However, performing de novo STRs calls on a genome-wide level from the existing WGS data is exceedingly demanding, both computationally and economically. Even if these calls were limited to “only” the ~200K UKB individuals not already analyzed by Halldorsson et al, we estimate this to last at least 40 days and incur costs well beyond EUR 25,000 (NB: these runs have to be performed remotely on dedicated UKB IT-infrastructure [UKB-RAP] with a specific cost plan). Unfortunately, this is beyond the financial capabilities of our group. For exactly these reasons, we had opted for analyzing the pre-computed STR data from Halldorsson et al. (2022) in the first place. We thank the reviewer and editor for their appreciation of the problem and their understanding!

Notwithstanding, the second part of the question, i.e. how imputed and WGS-derived STRs correspond, is highly relevant and addressable. To this end (and motivated by the comments of all reviewers), we have now substantially improved the relevant sections in the revised manuscript. For instance, we now report correlations for allele frequencies, effect size estimates (beta values), and sum of allele sizes between imputed and WGS-derived STRs not only for all n=84 previously manually curated STRs, but also for ~600K other STRs for which this was possible in the revised Results (section “Comparison of imputed vs. WGS-derived STRs” first paragraph, middle and second paragraph, beginning) and Methods (section “Comparison of GWAS results for imputed vs. WGS-derived STR variants”). For other improvements on this issue, please also see our responses to Comments 1.12 and 2.4 above. We hope that these extended comparisons and analyses across STR classes go in the direction that the reviewers had intended in their comment.

Comment 3.7: Sample size and power: The requirement to use only unrelated samples results in a loss of power due to reduced sample size. Current generalized mixed models used for incorporating related individuals in SNP data analyses (e.g., SAIGE and REGENIE) are not currently applicable to multi-allelic marker data, but this limitation should be discussed.

Response 3.7: We have followed the reviewer's suggestion of including this limitation into the discussion of the revised manuscript (section "Discussion").

Comment 3.8: Comparisons with previous work: The paper would benefit from a more detailed comparison with the Guo et al. (2025) study, including an analysis of concordance between results.

Response 3.8: We agree with this comment, which overlaps with a similar suggestion made by Reviewer 1.

In summary, the only STR variant reported to be genome-wide significant by Guo et al. (chr19:44921098), also shows genome-wide significance in our WGS-based data; indeed, with $p=2.99e-236$ it represents the third strongest signal overall. Furthermore, it is only 14bp apart from the most significant WGS-based variant in our data (chr19:44921083, $p=5.69e-303$). LD analysis shows strong correlation among both variants ($R^2=0.75$ and $D'\sim 1$) indicating that both STRs likely capture the same association signal. This detail was added to the relevant section in the revised manuscript (section "Discussion").

Comment 3.9: APOE region analysis: The authors highlight MHC signals in the discussion but provide less focus on the APOE signal despite analyzing it in depth. A graphic annotating the APOE region with all relevant markers, illustrating relative physical map positioning, LD (r^2), and types of STRs would be helpful, as one of the key messages is the increased complexity in the APOE region.

Response 3.9: While we generally agree that elaborating on the APOE region would be interesting, we have decided against it for two reasons: First, the main focus of our study was to identify *novel* loci potentially driven by STRs previously not highlighted by SNPs. APOE, of course, is the best and longest known AD risk locus and was recently also highlighted as a potential STR finding in the study by Guo et al. Therefore, it can be considered a "known and established" AD risk factor, both from the perspective of SNPs and STRs. Second, we note that detailed fine-mapping of association signals in this region is notoriously difficult (even for SNPs), owing to the strength of the underlying association

signals and the extensive LD structure. There are many SNP-based fine-mapping studies with partially contradictory findings in the literature displaying this complexity (e.g. Zhou et al. 2018, Kulminski et al. 2020, Martin et al. 2000). Hence, detailed fine-mapping weighing the effects of SNPs against STRs and vice versa would represent a major additional effort that would require and deserve an APOE-dedicated fine-mapping study, which is beyond the scope of the current analyses. We added abovementioned references to the revised manuscript (section “Discerning the drivers of STR-based GWAS signals”) and we sincerely hope the reviewer (and editor) understand this limitation!

Comment 3.10: Sex-stratified analysis: A separate male and female analysis should be conducted, in line with Nature Communications reporting guidelines.

Response 3.10: We thank the reviewer for this important comment! Motivated by this suggestion, we have now implemented this topic in two ways into the revised manuscript: First, we performed sex-stratified GWAS analyses to search for potential (and novel) sex-specific loci. Second, we ran a genome-wide interaction analysis using “genotype x sex” as interaction term to search for loci that show substantial association differences in males vs. females. In summary, neither of these analyses provided strong evidence for sex-specific effects in the primary GWAS signals, nor did they reveal any strong novel association signals in either males or females. Perhaps the only notable exception was observed within the MS4A6A association region on chromosome 11q12: This reached genome-wide significance in the male-only analyses ($P=1.83E-10$) but showed only nominal association in the female cohort ($P=0.005$), with the same effect direction in both subgroups. However, this difference was not significant in the interaction analyses after multiple testing correction and, thus, likely reflects a normal fluctuation of the association evidence. The results from these novel analyses (including MH plots for the sex-specific and genome-wide interaction analyses) have now been added to the revised manuscript in several places (Methods: “Genome-wide association analyses”; Results: “Genome-wide association analyses on imputed STRs”, Supplementary figures 26 and 27 and Supplementary Table 21).

Comment 3.11: Line by line revisions

Response 3.11: Please see below for line-by-line responses to these comments.

1. Line 127: The replication study should employ Bonferroni correction ($0.05/14$, not 0.05).

Response: We agree and have implemented this into the revised version. While this adjustment now reduced the number of variants showing direct evidence for replication, the improvement in the GWAS test statistics upon combining data from both analyses

remains unchanged. Overall, we still consider the amount of replication as noteworthy and supportive of our discovery findings.

2. Terminology: Replace "descent groups" with "ancestry groups" throughout the manuscript.

Response: Done.

3. P7 Line 143: Include the p-value for the COJO analysis of ABCA7 results.

Response: Done.

4. P7 Lines 149-155: Clarify whether these loci have been previously examined for multiple SNP signals, and whether conditioning took into account additional signals.

Response: We thank the reviewer for pointing this out. We have re-examined all loci reported in Bellenguez, 2022 overlapping with our genome-wide significant top STRs for additional SNP signals within 500kb. The only locus identified was chr8:27609395:G:GA (corresponding to SNP rs11787077, chr8:27607795, *CLU*), for which Bellenguez reported an independent secondary signal (rs73223431,chr8:27362470, *PTK2B*) located approximately 250kb from our lead variant. However, as conditioning on the lead SNP in this region (rs11787077) already accounted for nearly the full STR association signal (p after conditioning=5.59E-01), conditioning with the second SNP in this region will not provide any new information. More importantly: for the two STR variants showing residual association after conditioning (*MINDY2/ADAM10* and *ABCA7*), no additional signals were reported by Bellenguez et al.

5. P8 Line 168: Replace "dropped above suggestive significance" with "dropped, but were still showing suggestive association signals."

Response: Done.

6. P10 Line 250: Clarify what is meant by "matching variants" and provide more details on the WGS-derived and imputed STRs with $p < 1e-05$.

Response: We agree that this description of 'matching' variants warranted further elaboration. As this is in line with comments made by all reviewers regarding the comparison of imputed vs. WGS derived variants, we have now substantially revised this section in the manuscript (Results: "Comparison of imputed vs. WGS-derived STRs", Methods and answers 1.12, 2.4, 3.6).

7. P10 Line 250: Consider conducting concordance analyses with the results from Guo et al.

Response: This is in line with Comments 1.4 and 3.8. Motivated by all of these comments, we have now extended our description comparing both studies in the revised manuscript.

8. P10 Line 258: Define what is meant by "highlighted" STRs.

Response: We apologize for not having been clearer on this. Highlighted STRs in this context means STRs showing genome-wide significance in the main GWAS analyses. This has now been clarified in the revised manuscript.

9. P10 Lines 219-231: Include contingency tables illustrating these results in supplementary materials.

Response: We have now added this information to the supplementary material of the revised manuscript (Supplementary Table 22).

10. P16 Line 371: Correct "decent" to "descent".

Response: Thank you for spotting this typo – this has now been corrected and replaced by the term “ancestry” (Comment 3.11.2).

11. Figures 1 and 2: Describe in the legend how the AD and AD-by-proxy results were combined to generate single GWAS results.

Response: We would like to note that we did not run separate GWAS for AD and AD-by-proxy phenotypes. Instead, we ran a single analysis using our combined proxy-risk-score for AD (see methods: “Procedure for “proxy phenotyping”). We apologize for the unclear wording, which has been improved in the revised legend (now Figures 1A and 1B).

Reviewer #4 (Remarks to the Author):

We thank Reviewer #4 for their participation in the review process.

REVIEWERS' COMMENTS

Response: We thank the reviewers for their renewed efforts to assess our manuscript and the changes made in response to their original comments. We are grateful for their very positive evaluations of our revisions and are particularly thankful for their continued overall very positive appraisal of our work!

Below, we provide a point-by-point response to the novel remarks to the authors. We note that not all comments required revisions to the manuscript.

Reviewer #1 (Remarks to the Author):

Comment 1.0: *I appreciate the modifications. The revised manuscript shows substantial improvement over the initial submission. I have a few additional comments on reviewer responses and the updated results.*

Response 1.0: Thank you very much for your positive appraisal of the modifications made during the revision process.

Comment 1.1: *Please review the interpretation of the eQTM. The current results shown in Figure 3 indicate that the STR is linked to decreased DNAm, which in turn leads to lower SNX32 expression. However, DNAm sites are expected to silence gene expression, so the expected direction should be the opposite. The authors must try to resolve this ambiguity and adjust their mechanistic interpretation accordingly. If there is no additional evidence to support this effect, please acknowledge caution when interpreting such an unexpected direction of effect.*

Response 1.1: We appreciate this comment and agree that the effect directions observed may appear unexpected from a “canonical” viewpoint, i.e. the expectation that increased DNAm leads to decreased gene expression and vice versa. This canonical situation can be considered typical for CpGs mapping to the promoter region of genes. However, the CpG of interest here maps to the first intron of the associated *SNX32* transcript. There is a large body of literature which shows that increased (or decreased) gene-body methylation is often associated with increased (or decreased) levels of gene expression (e.g. PMIDs: 20133333, 34520760, 23325432, 19329998). Since this sequence of effect directions may appear counter-intuitive, we had referred to the *SNX32* finding of our study as a “non-canonical” effect. To make this clearer to potential readers of our manuscript, we have now revised our wording in the corresponding paragraph and have added the four abovementioned papers in support. We hope that the reviewer (and editor) agree that this is now phrased carefully enough to not warrant another revision at this stage. Specifically, we wrote:

The latter was between cg15531562 (*SNX32*, intron 1) and *SNX32* transcript ENST00000308342, showing a non-canonical effect direction, where decreased methylation in parts of the gene-body is associated with lower levels of gene expression, and vice-versa⁴²⁻⁴⁵. This is precisely what we observe in our study, i.e., a positive correlation between DNAm and *SNX32* mRNA expression (EC: $r=0.523$, $p=0.0047$; Figure 3A, Supplementary Data 16).

In case the reviewer (and editor) feel that this wording does still not sufficiently address the observed effect directions we would be happy to revise this further.

Comment 1.2: *The responses for the number of PCs used to adjust the eQTL and mQTL analysis may not adequately control for confounders. The authors should either: (a) demonstrate that their PC choices are optimal by testing a range of values and showing that their chosen numbers yield the best trade-off between power and control (e.g., as is done in GTEx analyses), or (b) explicitly acknowledge in the Methods and Results that the PC choices were not optimized and that this represents a limitation, particularly for the SNX32 interpretation.*

Response 1.2: We decided to implement option b) outlined by the reviewer and have now added the requested sentences to the Methods and Results of our re-revised version. Specifically, this now reads:

Methods [sections “Methylation quantitative trait locus (meQTL) analysis” and “Expression quantitative trait locus (eQTL) analysis”]: “Our choice of how many PCs to include in the model was based on scree plots made from the quantified mRNA data. We did not test combinations with other numbers of PCs for this analysis, which represents a potential limitation of this arm of our study.”

Results [section “Delineating potential functional mechanisms of AD-associated STRs”]: “As stated in the Methods, we did not test combinations using other numbers of PCs in the eQTL/meQTL analysis model, which represents a potential limitation of this arm of our study.”

Comment 1.3: *In Response 1.17, the authors address my concern about the robustness of heritability estimates derived from random subsets by rerunning the analysis across 17 independent batches. They report a mean STR heritability estimate of 2.6% but acknowledge that “heritability estimates from each run varied.” I ask the authors to report the range and/or standard deviation of the 17 estimates in the Results section, in addition to the mean. If the variance across runs is large relative to the mean estimate, this should be discussed as a limitation.*

Response 1.3: We thank the reviewer for raising this point. In our revised submission we had fully disclosed the heritability estimates for each of the 17 batches. The standard deviation across these computations is 2.98%, which is, indeed, comparatively large relative to the mean (~2.6%). This indicates that STR heritability calculations in this context are sensitive to the specific subsets of individuals analyzed and should be interpreted with caution. One possible reason for this variance lies in the formula used to calculate the relative contributions: Although the absolute heritability estimates for both the combined and SNP-only models remain stable across subsets (standard deviation < 10% of the respective means), the absolute difference between them is inherently very small. Dividing this difference by the total heritability mathematically amplifies normal fluctuations, leading to the comparatively high variation observed in our study.

We have updated the respective section in the manuscript to reflect this. Specifically, we wrote (Results [section “Heritability estimates for STRs compared to SNPs”]):

We note that the standard deviation of the heritability estimates across all 17 batches is 2.98%, and therefore comparatively large relative to the mean. One possible reason for the high variance lies in the small relative contribution made by the STRs to the absolute heritability. As a result, the contribution metric becomes sensitive to minor natural variations between the different subsets and should,

therefore, be interpreted with some caution.

Reviewer #2 (Remarks to the Author):

Comment 2.0: *The authors have adequately addressed all of my concerns.*

Response 2.0: We thank the reviewer for his/her response to our modifications and appreciate the positive feedback.

Reviewer #3 (Remarks to the Author):

Comment 3.0: *The authors have substantially improved reporting of effect sizes and standard errors for conditional analyses at key loci (ABCA7, MINDY2/ADAM10), now presented both in-text and via new main-text figures (Figures 2C, 2D). The expanded concordance analysis between imputed and WGS-derived STRs which now covers ~622,000 STRs rather than only the 84 manually curated variants provides additional evidence that the imputation approach is sound. Moreover, the sensitivity analyses for age adjustment and proxy-case adequately demonstrate robustness to these analytical choices. The Bonferroni correction for replication ($p < 0.05/14$) has been appropriately applied.*

The targeted multi-allelic STR analyses, performed in response to concerns about information loss from biallelic coding, confirmed 12 of 14 genome-wide significant loci, strengthening confidence in the primary findings. The sex-stratified GWAS and genotype-by-sex interaction analyses found no strong evidence for sex-specific effects satisfying our request. The concordance with Guo et al. at the APOE locus ($r^2 = 0.75$, $D' \approx 1$ between their respective lead STRs, separated by only 14bp) is reassuring. We accept the authors' justification for not producing a detailed APOE fine-mapping figure as this is probably out of scope for this manuscript.

The analysis code is now available on GitHub and summary statistics have been deposited on Zenodo as requested in our original review.

The remaining limitations, specifically the reliance on imputed STRs, restriction to European-ancestry samples and small functional validation sample, were acknowledged by the authors in the review comments and don't detract from the overall results.

Response 3.0: We thank this reviewer for the re-evaluation of our manuscript and are happy to read that they are happy with the outlined modifications, which we made in response to the original review. We also believe that our manuscript has been strengthened by implementing these changes, so thank you again for raising these points in the first place!

Comment 3.1: *The claim of inability to perform de novo STR calling from WGS to compare the imputed STRs to due to computational cost (Rebuttal point 3.6) requires a counterpoint. The authors claim about cost suggest that they thought that an evaluation across all STRs and individuals is required for this, but that is not the case. This could be performed with a random sample of 100 individuals across 100 randomly chosen loci. If the authors are not familiar with the tools or working in RAP this would have*

been a hurdle from an implementation point of view, but it is not a cost hurdle, as the authors claim, especially since UKBB RAP give compute credit in the first instance anyway. However, whilst desirable there is enough other supporting evidence, especially in the revision, and previous work, to render this not a strict requirement and the paper is acceptable in its current state.

Response 3.1: We apologize for the misunderstanding in our first response letter. We agree with this reviewer's notion that addressing this point is no longer a strict requirement given the other sources of supporting evidence. Hence, we have not made any changes to the re-revised manuscript.

Reviewer #4 (Remarks to the Author):

Response: As before, we thank this reviewer for their participation in the review process.